# Exponential Scaling of Factual Inconsistency in Data-to-Text Generation with Fine-Tuned LLMs

**Joy Mahapatra**  *joymahapatra90@gmail.com*
*Indian Statistical Institute Kolkata*

**Soumyajit Roy**  *soumyajitroy25356@gmail.com*
*Indian Statistical Institute Kolkata*

**Utpal Garain**  *utpal.garain@gmail.com*
*Indian Statistical Institute Kolkata*

**Reviewed on OpenReview:** *https://openreview.net/forum?id=xPaPd6g5WG*

## Abstract

Data-to-text (D2T) generation is a core task in text generation that involves converting semi-structured data (e.g., tables, graphs) into text. Recent advances in large language models (LLMs) have led to significant improvements in D2T. Despite these gains, factual inconsistency remains a persistent issue in LLMs for D2T. Understanding how such inconsistencies scale with factors like model size, compute (FLOPs), and data size is crucial for building trustworthy systems. While prior scaling studies focus on generalization error via power law scaling, the impact of these factors on factual inconsistency in D2T remains unexplored. This paper addresses the gap by empirically investigating how factual inconsistency scales with various scaling factors. Unlike prior studies that focus solely on power law scaling, we also examine exponential scaling. To rigorously compare these models, we introduce *VaCScal*, a three-stage statistical validation framework: (1) predictive performance estimation, (2) goodness-of-fit assessment, and (3) comparative analysis. Experiments are conducted across six diverse LLM families and five D2T datasets. Factual inconsistency is inversely measured using four state-of-the-art consistency metrics, including human evaluation. We employ QLoRA, Prefix-Tuning, and full fine-tuning to fine-tune the LLMs. Our analysis, validated through the *VaCScal* framework, consistently shows that factual inconsistency in D2T generation follows exponential scaling with respect to model (LLM) size, compute (FLOPs), and fine-tuning data size—challenging the prevailing assumption of power law scaling. To support this finding, a mathematical rationale is also provided, demonstrating why exponential scaling behavior is expected in factual inconsistency under typical D2T conditions.

## 1 Introduction

Data-to-text (D2T) generation (Lin et al., 2024b; Li et al., 2024) involves converting semi-structured data (e.g., tables, graphs) into natural language, with applications in dialogue systems, automated journalism, and beyond. LLMs have achieved strong performance on D2T tasks, particularly in coherence and informativeness (Lorandi & Belz, 2024). However, factual inconsistency—where the generated text fails to faithfully reflect the input data—remains a major challenge (Li et al., 2022; Huang et al., 2023), often diminishing trust in LLM-based D2T systems (Figure 1). To build reliable models, it is crucial to understand how factual inconsistency scales with key factors such as model size, compute (FLOPs), and training (fine-tuning) data. Nonetheless, existing scaling analyses in D2T focus almost entirely on generalization errors or test perplexity, especially under power law scaling (Kaplan et al., 2020; Hoffmann et al., 2022), while overlooking

how factual inconsistency behaves under similar scaling. Bridging this gap is essential for advancing both research understanding and the practical development of more trustworthy D2T systems.

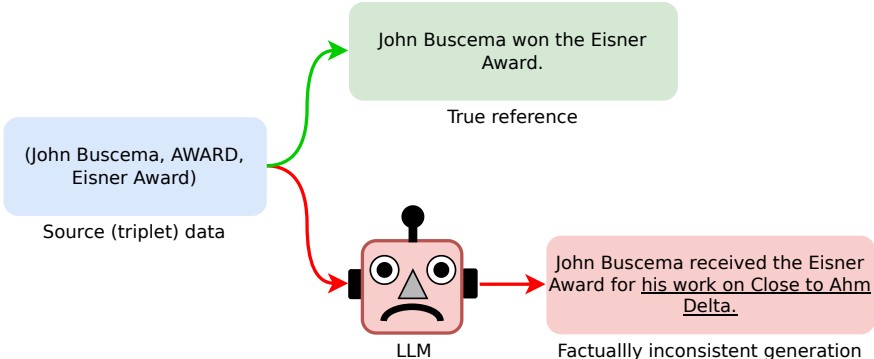

Figure 1: An example from the DART (Nan et al., 2021) dataset demonstrating a factually inconsistent output generated by an LLM (Pythia-1.4B). The generated text contains unsupported content—*"his work on Close to Ahm Delta"*—which is not grounded in the source data.

In this paper, we address a critical research gap by examining how factual inconsistency scales in D2T generation. Unlike prior work focused solely on power law scaling (Kaplan et al., 2020; Hoffmann et al., 2022; Brandfonbrener et al., 2025), we also explore exponential scaling using *VaCScal*—a three-stage statistical validation framework consisting of: (1) predictive performance estimation, (2) goodness-of-fit assessment, and (3) comparative analysis. *VaCScal* enables rigorous and principled comparison between power law and exponential scaling behaviours. Our study covers three major LLM architectures (decoder-only, encoder-decoder, and state-space), evaluating six LLM families—BLOOM, FLAN-T5, Mamba, OPT, Pythia, and Qwen2.5—across five D2T datasets: E2E, ViGGO, WebNLG, DART, and WikiTableText. Models are fine-tuned using QLoRA (Dettmers et al., 2023) and Prefix-Tuning (Li & Liang, 2021). To ensure broader adoption, we complement these fine-tuning approaches with experiments involving full fine-tuning, which updates all trainable model parameters. For robustness toward decoding, we report results using both greedy and nucleus sampling decoding strategies. Factual inconsistency is defined as the inverse of factual consistency and measured using four automatic metrics: AlignScore, QAFactEval, SummaC-conv, and UniEval-fact, along with human annotations. Our findings, validated through *VaCScal*, reveal that factual inconsistency exhibits exponential scaling across multiple factors—such as model size, compute, and fine-tuning data—challenging the dominant assumption of power law scaling. To support these empirical results, we also provide a mathematical rationale showing that factual inconsistency is expected to scale exponentially with LLM size under typical D2T conditions.

## 2 Related Work

**Data-to-text generation (D2T) and factual inconsistency.** Data-to-text (D2T) generation (Lin et al., 2024b) involves generating natural language from structured inputs such as graphs (Gardent et al., 2017; Nan et al., 2021), tables (Bao et al., 2018), and meaning representations (slot-value pairs) (Novikova et al., 2017; Juraska et al., 2019). These three representation forms constitute the main categories of D2T: graph-to-text, table-to-text, and MR-to-text Recently, LLMs have become central to D2T due to their large-scale pretraining (Zhang et al., 2022), scalability (Scao et al., 2022), and compatibility with efficient fine-tuning (Dettmers et al., 2023) and prompt-based learning (Lester et al., 2021). These models often outperform traditional approaches in coherence and informativeness (Ge et al., 2023). However, factual inconsistency—where generated text fails to reflect input data accurately—remains a persistent challenge (Fabbri et al., 2022; Zha et al., 2023), leading to hallucinations and reduced trustworthiness. Prior work attributes this to misalignment between source data and references (Dhingra et al., 2019; ul Islam et al., 2023), LLM biases (Zhang et al., 2023), and exposure bias (Huang et al., 2023). While human evaluation is the gold standard, recent

automatic metrics (Fabbri et al., 2022; Zha et al., 2023) have shown strong correlation with human judgments and offer scalable alternatives.

**Scaling in LLM.** In LLMs, *scaling* describes how LLM performance varies with scaling factors such as model size, compute, data, etc (Chung et al., 2024; Zhang et al., 2024; Brandfonbrener et al., 2025; Mayil-vahanan et al., 2025). Hestness et al. (2017) showed that deep language models follow power law scaling, which was extended by Kaplan et al. (2020) to include model size, dataset size, and compute. Subsequent works have explored scaling across tasks (Bansal et al., 2022; Kaplan et al., 2020), paradigms such as sparse modeling (Frantar et al., 2024), and parameter-efficient fine-tuning (Zhang et al., 2024). More recently, joint scaling—such as additive and multiplicative formulations—has garnered increasing attention (Hoffmann et al., 2022; Zhang et al., 2024). Scaling plays a critical role in hyperparameter tuning (Hendrycks, forthcoming), training cost estimation (Hägele et al., 2024), and model performance prediction (Hoffmann et al., 2022). A set of recent studies (Atanasov et al., 2024; Bahri et al., 2024; Lin et al., 2024a) has further reinforced the theoretical foundations of scaling.

## 3 Desiderata: Scaling Models

Moving beyond prior work on power law scaling (Kaplan et al., 2020; Brandfonbrener et al., 2025; Gadre et al., 2025), we examine two scaling models for factual inconsistency in D2T generation with respect to LLM size: power law scaling (modeled by a power law function) and exponential scaling (modeled by an exponential function). The two scaling models are defined as follows:

$$\text{Power Law Scaling Model} : f(x) = \begin{cases} Ax^\alpha + B & x \geq 0 \\ 0 & \text{otherwise} \end{cases} \tag{1}$$

$$\text{Exponential Scaling Model} : f(x) = \begin{cases} Ce^{\beta x} + D & x \geq 0 \\ 0 & \text{otherwise} \end{cases} \tag{2}$$

Here, $A$ and $C$ are case-specific parameters, $\alpha$ is the power law exponent, $\beta$ is the exponential scaling rate, $x$ represents the scaling factor (e.g., LLM size, compute (FLOPs), etc.), and $f(x)$ denotes the scaling objective (e.g., factual inconsistency). $B$ and $D$ represent the irreducible error or entropy of an LLM family (Brandfonbrener et al., 2025); in other studies, they are interpreted as the performance threshold or capacity of a given LLM family (Hestness et al., 2017). Following the significant work by Zhang et al. (2024), we estimate the parameters of both models via maximum likelihood estimation on the factual inconsistency data $\mathcal{D}$, optimizing with the standard Huber loss ($\delta = 1$) due to its robustness to outliers.

## 4 VaCScal: Statistical Validation Framework

Prior empirical studies on LLM scaling often assess predictive performance using held-out loss alone. However, we find this insufficient for rigorously evaluating or comparing multiple scaling models. To address this, we use *VaCScal* (Validation and Comparison of Scaling law)—a structured three-stage framework comprising predictive performance estimation, goodness-of-fit assessment, and comparative analysis. Each stage is detailed below.

- **Stage I: Predictive Performance Estimation.** The first stage evaluates how well the scaling models generalize in terms of their predictive ability on unseen data. To achieve this, we assess the scaling models on held-out data using the Huber loss ($\delta = 1$). The Huber loss is known for its robustness to outliers, making it well-suited for reliable estimation. Given the limited data availability, we employ five-fold cross-validation to assess predictive performance.

- **Stage II: Goodness-of-fit Assessment.** Predictive performance alone is not sufficient to validate a scaling model; assessing its goodness-of-fit is also crucial for its acceptance. Therefore, in this

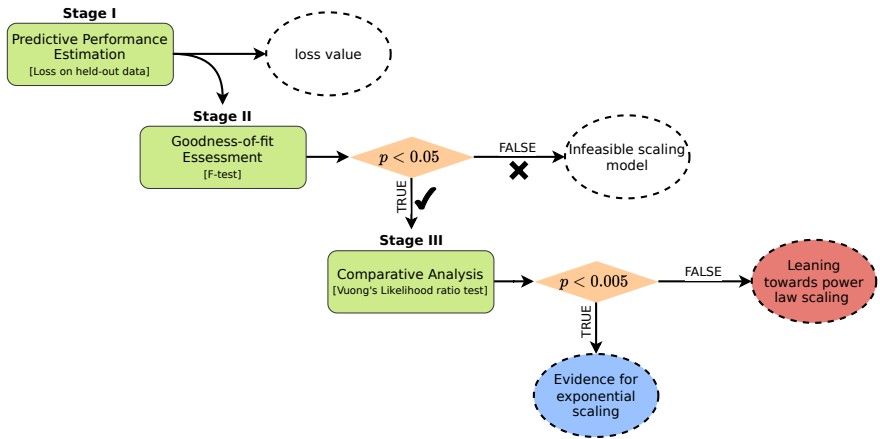

Figure 2: All three stages of *VaCScal*, the statistical validation framework for scaling models. Dashed ovals indicate outputs of each stage.

stage, we assess the goodness-of-fit of the scaling models using an F-test for regression (Weisberg, 2005; Siegel, 2016). The test statistic for the F-test is computed as follows:

$$F_{\text{stat}} = \frac{SSR_{\mathfrak{R}} - SSR_{\mathfrak{E}}}{df_{\mathfrak{R}} - df_{\mathfrak{E}}} \bigg/ \frac{SSR_{\mathfrak{E}}}{df_{\mathfrak{E}}}$$

$$F_{\text{stat}} \sim \text{F-distribution}(x)$$

Here, $SSR_{\mathfrak{R}}$ and $SSR_{\mathfrak{E}}$ represent the sum of squared residuals for the reduced and exact models, respectively. Similarly, $df_{\mathfrak{R}}$ and $df_{\mathfrak{E}}$ denote the degrees of freedom for the reduced and exact models, respectively. We consider both of our scaling models as exact models, while the reduced model is represented by a simple mean-response model. Since the F-test applies only to linear regression models, we use a log transformation to convert our scaling models into their linear forms. We perform the F-test with a significance level of $p < 0.05$, which is commonly considered a moderate threshold. If a scaling model fails to achieve a $p$-value below 0.05 in the goodness-of-fit assessment, we do not consider it feasible for modeling scaling behavior.

- **Stage III: Comparative analysis.** In this final stage of validation, we compare the two scaling models through hypothesis testing to determine which better explains the data. Since power law and exponential scaling models are not nested hypotheses, the standard likelihood-ratio test is not applicable. Instead, we employ Vuong's likelihood-ratio test (Vuong, 1989) for comparison. The test statistic for Vuong's likelihood-ratio test is computed as follows:

$$V_{\text{stat}} = \frac{\sqrt{n} \cdot \text{mean}(d)}{\sqrt{\text{Var}(d)}} \tag{3}$$

$$V_{\text{stat}} \sim \text{normal}(0, 1) \tag{4}$$

Where $n$ represents the sample size, and $d$ denotes the $n$-sized sample of the log-likelihood differences between the two scaling models. We conduct Vuong's likelihood ratio test at a stringent significance level of $p < 0.005$ to provide highly compelling evidence for our conclusion.

**Normality Assumptions Verification.**  In Stage II and III of our validation framework, we incorporate the F-test and Vuong's likelihood-ratio test. Both tests rely, directly or indirectly, on the assumption that the residuals of our scaling models follow a normal distribution. Therefore, validating this assumption is essential. To assess the normality of residuals for both scaling models, we employ the Shapiro–Wilk test (Shaphiro & Wilk, 1965) (with $p < 0.05$) throughout all of our experiments.

# 5 Experiment Setup

We provide brief details about the LLM families, D2T datasets, and metrics used to measure factual inconsistency. For additional information on settings and other critical implementation details, please see Appendix A. All code used in our scaling experiments is provided in the Supplementary Material.

## 5.1 D2T Datasets and LLM Families

Our experiments incorporate five widely used D2T datasets, covering three major D2T generation types: DART (Nan et al., 2021) and WebNLG (Gardent et al., 2017) for graph-to-text, WikiTableText (Bao et al., 2018) for table-to-text, and E2E (Dusek et al., 2018) and ViGGO (Juraska et al., 2018) for MR-to-text. Among these five D2T datasets, DART (represents diverse knowledge triples), WebNLG (extracted from DBpedia), and WikiTableText (extracted from Wikipedia) are open-domain datasets that capture broad knowledge, whereas E2E (restaurant domain) and ViGGO (video game domain) are closed-domain datasets that represent consistent, domain-specific knowledge. Additional details about these datasets are provided in Appendix A.1. We also evaluate six prominent LLM families spanning three architectural paradigms: decoder-only, encoder–decoder, and state-space. These include BLOOM (5 models) (Scao et al., 2022), FLAN-T5 (5 models) (Chung et al., 2024), OPT (6 models) (Zhang et al., 2022), Pythia (8 models) (Biderman et al., 2023), Mamba (5 models) (Gu & Dao, 2024), and Qwen2.5 (7 models) (Yang et al., 2024). Further details are available in Appendix A.2. All datasets and model variants are sourced from the Hugging Face hub (Wolf et al., 2020). A summary of the LLM families is presented in Table 1.

| Family | Model Paradigm | Model Counts | Parameters of Each Models |
|---|---|---|---|
| BLOOM | decoder-only | 5 | 560M, 1.1B, 1.7B, 3B, 7B |
| FLAN-T5 | encoder-decoder | 5 | 77M, 248M, 783M, 2.85B, 11.3B |
| Mamba | state-space model | 5 | 130M, 370M, 790M, 1.4B, 2.8B |
| OPT | decoder-only | 6 | 125M, 350M, 1.3B, 2.7B, 6.7B, 13B |
| Pythia | decoder-only | 8 | 70M, 160M, 410M, 1B, 1.4B, 2.8B, 6.9B, 12B |
| Qwen2.5 | decoder-only | 7 | 0.5B, 1.5B, 3B, 7B, 14B, 32B, 72B |

Table 1: Overview of the five LLM families, their architectural paradigms, and model sizes (M = million, B = billion parameters).

## 5.2 Settings

**Input Linearisation, Prompt Structure, Fine-tuning, and Decoding Strategies.** In D2T tasks, input linearisation involves transforming semi-structured data into a sequential representation before feeding it into an LLM. We consider two forms of linearisation: (1) lists of slot–value pairs and (2) lists of relational triples, as detailed in Appendix A.5. For supervised fine-tuning across all our experiments, we employ a description-guided prompt structure, also provided in Appendix A.5. We use two parameter-efficient fine-tuning strategies—QLoRA (Dettmers et al., 2023) and Prefix-Tuning (Li & Liang, 2021)—to fine-tune all LLMs on each D2T dataset. Alongside these two parameter-efficient fine-tuning methods, we also perform full fine-tuning to broaden the reach of our results. For more details on the fine-tuning setup, please see Appendix A.3. For decoding, we consider both greedy decoding and nucleus sampling. For further training settings and library details, please refer to Appendix A.4.

**Quantification for Factual Inconsistency.** We define factual inconsistency as the inverse of factual consistency, computed as $(1 - z)$, where $z$ is the factual consistency score ranging from 0 to 1. To evaluate factual inconsistency in LLMs for D2T tasks, we employ four state-of-the-art automatic metrics that exhibit strong correlations with human judgments: ALIGNSCORE, which measures consistency through information alignment (Zha et al., 2023); QAFACTEVAL, which assesses consistency via question generation and answering (Fabbri et al., 2022); SUMMAC-CONV, which leverages natural language inference techniques (Laban et al., 2022); and UNIEVAL-FACT, which utilizes a unified evaluation framework based on multi-task train-

ing (Zhong et al., 2022). In addition to these automatic metrics, we also conduct human evaluation, wherein factual consistency is manually annotated by human annotators.

# 6 Results

This section presents the plots and *VaCScal* results for factual inconsistency scaling (measure using ALIGN-SCORE and QAFACTEVAL) in D2T with respect to two scaling factor model size and compute (FLOPs), based on two scaling models—power law and exponential. All results in this study are based on text generated using greedy decoding.

## 6.1 Scaling Behavior Based on Model Size of Fine-Tuned (QLoRA) LLM Families

Here, we present the scaling behavior of factual inconsistency in D2T with respect to model size for LLM families fine-tuned using QLoRA. Here, we present results for four LLM families—FLAN-T5, Mamba, Pythia, and Qwen2.5. Results for the other two families, BLOOM and OPT, are provided in Appendix C. For results using SUMMAC-CONV and UNIEVAL-FACT as factual inconsistency metrics, please refer to Appendix D.

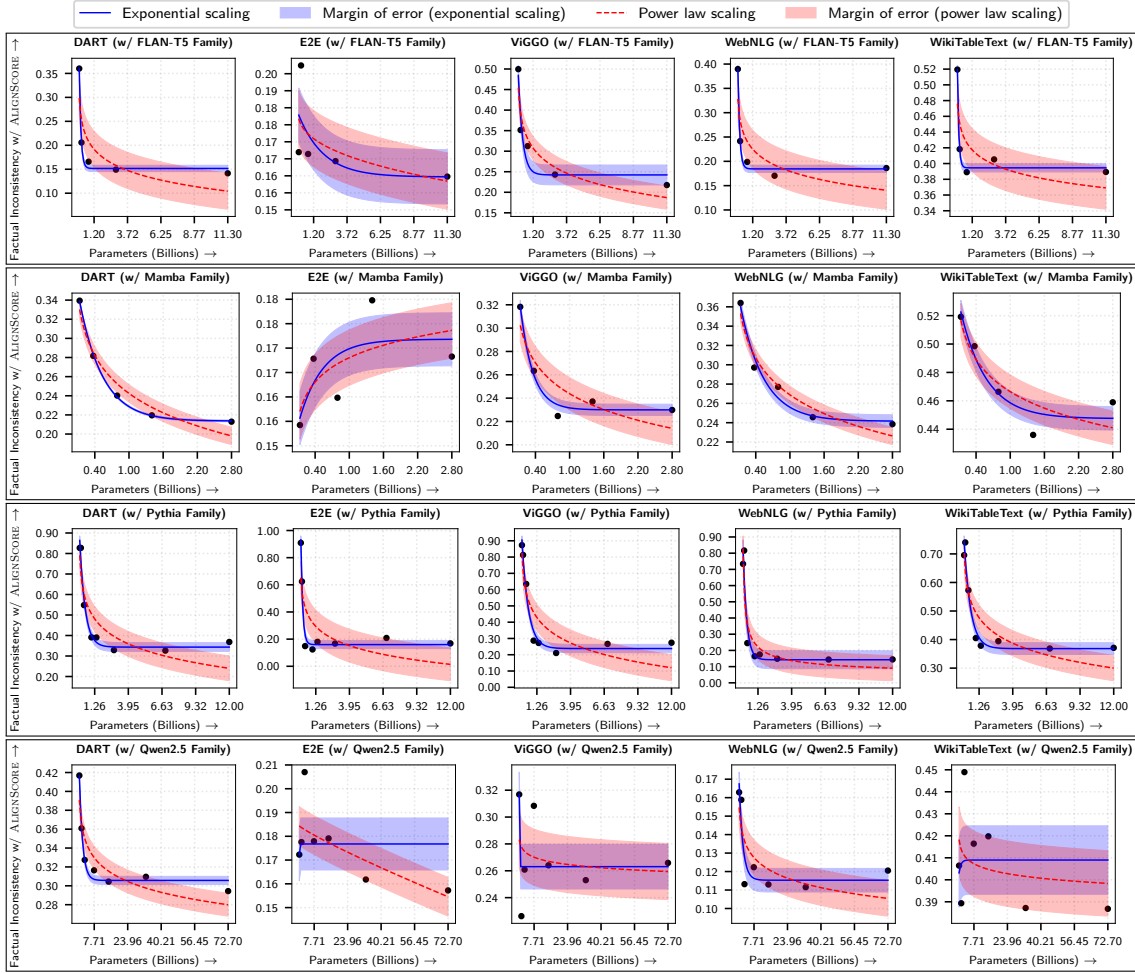

Figure 3: Fitted power law and exponential scaling models (with 95% confidence intervals) for factual inconsistency, measured by ALIGNSCORE, plotted against model size across different LLM families.

### 6.1.1 Factual Inconsistency Measured via AlignScore

Figure 3 shows the fitted curves for the power law and exponential scaling models, along with 95% confidence intervals derived from residual error. Anomalies are particularly observed in the E2E dataset for the Mamba family, where factual inconsistency unexpectedly increases with model size. A possible hypothesis for this aberrant behavior is discussed in Appendix I. In sporadic exceptional cases involving the E2E and ViGGO datasets, both scaling models exhibit higher margins of error (MoE), although the exponential model consistently shows lower MoE than the power law. Outside these exceptions, exponential scaling generally yields much narrower MoEs, indicating more stable and reliable fits in terms of uncertainty. Table 2 summarizes the outcomes of the *VaCScal* framework. Exponential scaling consistently outperforms the power law across all three validation stages: in Stage I, it achieves lower held-out loss; in Stage II, it passes the F-test more frequently (✔); and in Stage III, Vuong's test favors exponential scaling in most cases (highlighted in blue). In several scenarios, power law scaling is found to be infeasible. Although in multiple cases for Qwen2.5 both scaling models fail to pass the Stage II feasibility test of *VaCScal* (i.e., meeting the minimum $p$-value for goodness-of-fit), exponential scaling still achieves lower or comparable held-out loss (Stage I) in most scenarios. Despite a few exceptions (e.g., E2E), *VaCScal* results indicate that exponential scaling more reliably models factual inconsistency—measured via ALIGNSCORE—in D2T generation.

**Takeaway.** *Exponential scaling models more accurately capture factual inconsistency (measured via ALIGN-SCORE) with respect to LLM model size in D2T, exhibiting lower error and stronger statistical fit compared to power law scaling, despite a few outliers.*

| LLM family | Scaling | DART | E2E | ViGGO | WebNLG | WikiTableText |
|---|---|---|---|---|---|---|
| FLAN-T5 | Exponential | 4.15E-04/✔ | 8.60E-04/✘ | 1.14E-03/✔ | 2.16E-03/✔ | 2.56E-03/✔ |
| | Power Law | 1.47E-04/✔ | 3.11E-03/✘ | 7.80E-01/✘ | 6.16E-04/✔ | 1.45E-04/✔ |
| Mamba | Exponential | 2.14E-05/✔ | 1.43E-04/✘ | 4.40E+01/✔ | 1.05E-04/✔ | 3.09E-04/✘ |
| | Power Law | 3.77E+05/✘ | 1.99E+03/✘ | 1.08E+167/✘ | 3.50E-04/✘ | 1.42E+13/✘ |
| Pythia | Exponential | 2.03E-03/✔ | 1.71E+01/✔ | 2.48E-03/✔ | 2.30E-02/✔ | 3.31E-03/✔ |
| | Power Law | 1.68E+11/✘ | 1.24E-02/✔ | 1.92E-02/✘ | 8.71E+11/✘ | 1.96E+04/✘ |
| Qwen2.5 | Exponential | 4.93E-05/✔ | 2.30E-04/✘ | 3.98E-02/✘ | 7.73E+01/✔ | 3.41E-04/✘ |
| | Power Law | 4.79E-01/✔ | 2.00E-03/✘ | 6.81E-04/✘ | 2.71E-04/✘ | 2.59E-03/✘ |

Table 2: Results from *VaCScal* corresponding to Figure 3. Numerical values, along with (✔/✘) and blue/red highlights, indicate the outcomes of Stages I, II, and III of *VaCScal*.

### 6.1.2 Factual Inconsistency Measured via QAFactEval

Figure 4 shows the fitted power law and exponential scaling curves, along with 95% MoE, using QAFACTE-VAL as the inconsistency metric. Similar to ALIGNSCORE, notable anomalies are observed—especially for the E2E dataset, where all LLM families (except Pythia) exhibit increasing inconsistency with model size. Aside from a few exceptions in E2E and ViGGO, where both scaling models show higher MoE, exponential scaling generally yields narrower MoE than power law scaling, indicating more stable fits with lower uncertainty. Table 3 summarizes the corresponding results under the *VaCScal* framework. In most Stage I results, exponential models consistently achieve lower or comparable held-out loss. For Stage II, they pass the F-test (✔) more reliably than power law models, which often fail (✘). Overall, Stage III also favors exponential scaling, with numerous significant Vuong test results (highlighted in blue). In two rare cases—DART and WebNLG with the FLAN-T5 family—power law scaling outperforms exponential models. We hypothesize that FLAN-T5, having been pre-trained on graph-to-text formats, may exhibit divergence from exponential scaling in these cases due to prior alignment with power law behavior. In Qwen2.5, apart from ViGGO, all other datasets show significantly lower held-out loss (Stage I) for exponential scaling. Overall, aside from these few rare exceptions, exponential scaling emerges as a more robust and statistically valid model of factual inconsistency in D2T when measured with QAFACTEVAL.

**Takeaway.** *Compared to power law scaling, exponential scaling more effectively captures the relationship between factual inconsistency (measured via QAFACTEVAL) and model size in D2T across most LLM families, despite some anomalies (e.g., E2E, ViGGO).*

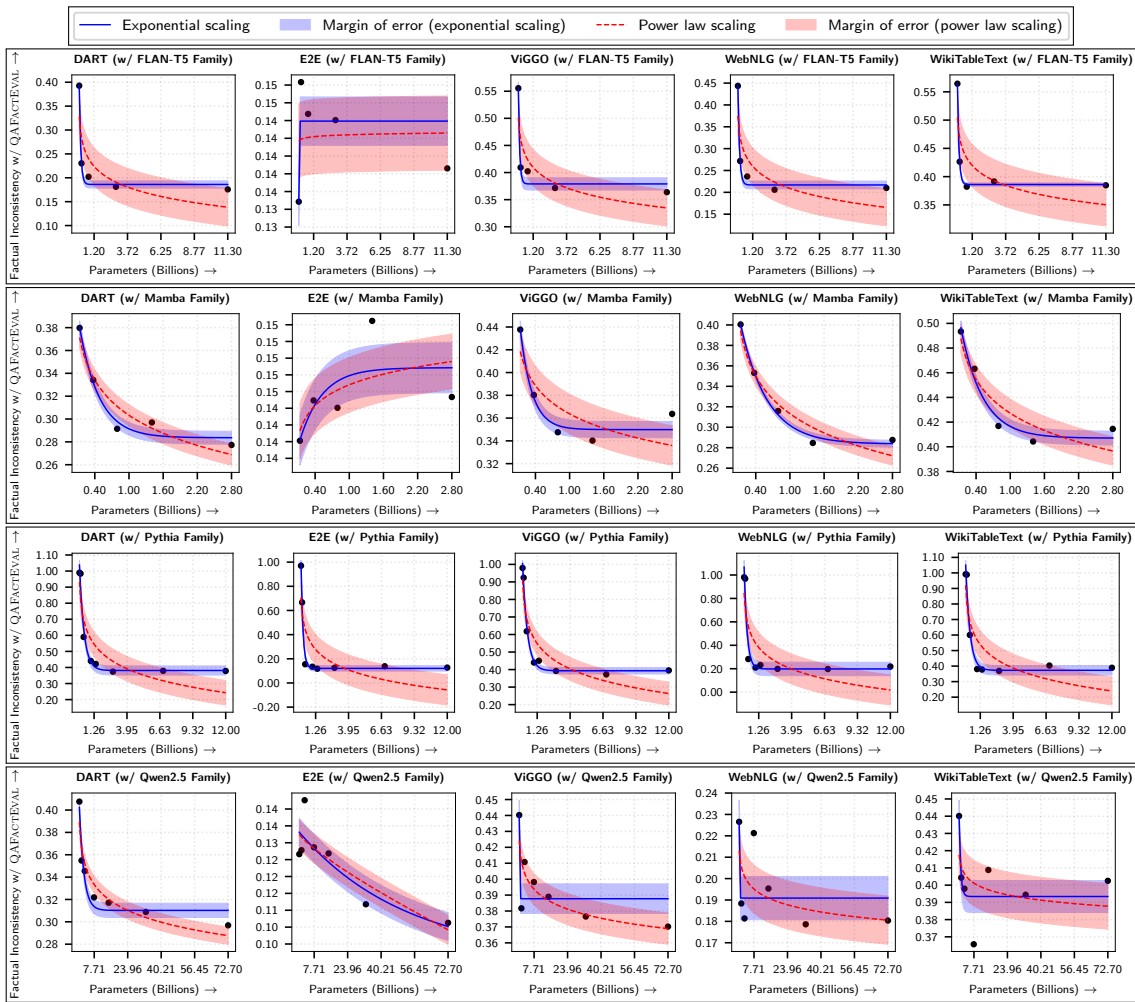

Figure 4: Fitted power law and exponential scaling models (with 95% confidence intervals) for factual inconsistency, measured by QAFACTEVAL, plotted against model size across different LLM families.

| LLM Family | Scaling | DART | E2E | ViGGO | WebNLG | WikiTableText |
|---|---|---|---|---|---|---|
| FLAN-T5 | Exponential | 3.37E-03/✔ | 6.59E-05/✘ | 4.64E-03/✔ | 2.46E-03/✔ | 3.22E+00/✔ |
| | Power Law | 1.22E+02/✔ | 7.08E-04/✘ | 1.30E-03/✔ | 2.57E-03/✔ | 4.57E-04/✔ |
| Mamba | Exponential | 3.74E-04/✔ | 3.75E-05/✘ | 1.52E+01/✔ | 4.38E-05/✔ | 5.68E-05/✔ |
| | Power Law | 4.01E-05/✘ | 2.17E-05/✘ | 4.43E-04/✘ | 2.63E+05/✘ | 9.08E+20/✘ |
| Pythia | Exponential | 3.97E-03/✔ | 7.47E-02/✔ | 3.65E-03/✔ | 4.54E-02/✔ | 3.24E-01/✔ |
| | Power Law | 1.04E+06/✘ | 2.86E-02/✔ | 2.09E+06/✔ | 9.44E+14/✘ | 1.73E-02/✘ |
| Qwen2.5 | Exponential | 3.26E-04/✔ | 7.24E-05/✔ | 4.64E-04/✘ | 2.06E-04/✘ | 2.88E-04/✘ |
| | Power Law | 1.06E-01/✔ | 2.14E-03/✘ | 3.63E-02/✔ | 2.91E-04/✘ | 6.27E+16/✘ |

Table 3: Results from *VaCScal* corresponding to Figure 4. Numerical values, along with (✔/✘) and blue/red highlights, indicate the outcomes of Stages I, II, and III of *VaCScal*.

## 6.2 Scaling Behaviour Based on FLOPs of Fine-Tuned (QLoRA) LLM Families

Estimating compute budget is central to LLM scaling research (Kaplan et al., 2020; Chung et al., 2024), with several early works linking generalization error to FLOPs (floating-point operations). This section examines the scaling behavior of LLM factual inconsistency in D2T with respect to FLOPs. We evaluate LLM families across five D2T datasets using two automatic metrics—ALIGNSCORE and QAFACTEVAL. Here, we present results for four LLM families—FLAN-T5, Mamba, Pythia, and Qwen2.5. Results for the other two families,

BLOOM and OPT, are provided in Appendix C. FLOPs are computed using `calflops`[1], and all models are fine-tuned using QLoRA.

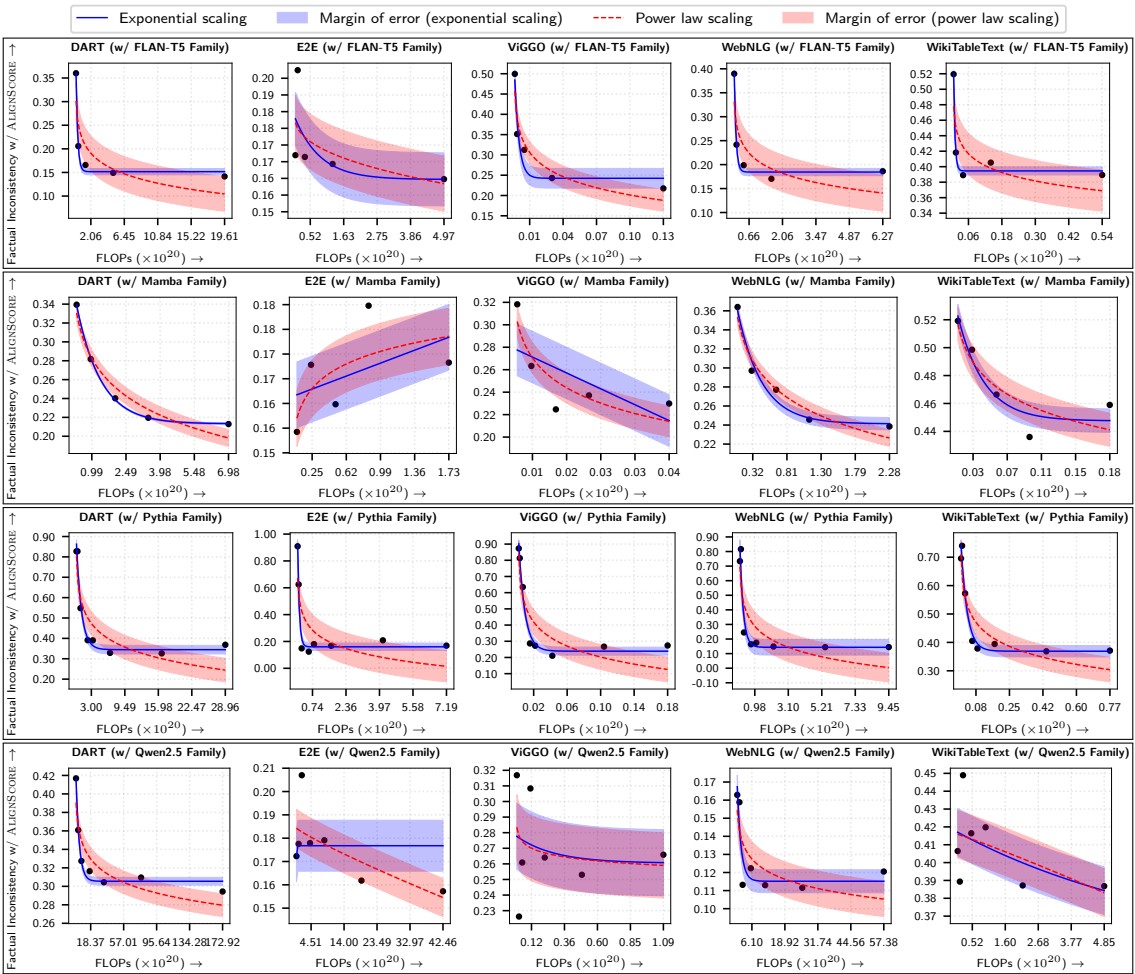

Figure 5: Fitted power law and exponential scaling models (with 95% confidence intervals) for factual inconsistency, measured by ALIGNSCORE, plotted against FLOPs across different LLM families.

| LLM Family | Scaling | DART | E2E | ViGGO | WebNLG | WikiTableText |
|---|---|---|---|---|---|---|
| FLAN-T5 | Exponential | 1.52E-03/✔ | 4.62E-04/✖ | 3.28E-03/✖ | 4.48E-04/✔ | 3.88E-03/✔ |
| | Power Law | 6.14E-05/✔ | 6.15E-03/✖ | 3.47E-01/✖ | 2.36E-03/✔ | 1.33E-04/✔ |
| Mamba | Exponential | 1.62E-05/✔ | 2.36E-04/✖ | 8.04E+01/✔ | 3.92E-04/✔ | 4.56E-04/✖ |
| | Power Law | 4.51E+05/✖ | 1.47E-03/✖ | 5.07E+176/✖ | 1.13E+00/✖ | 4.44E+13/✖ |
| Pythia | Exponential | 3.82E-03/✖ | 3.22E+01/✔ | 1.24E-03/✔ | 4.92E-02/✔ | 1.03E-03/✔ |
| | Power Law | 1.95E+11/✖ | 2.56E-02/✔ | 2.48E+03/✖ | 1.77E+11/✖ | 1.37E+03/✖ |
| Qwen2.5 | Exponential | 9.97E-05/✔ | 1.76E-04/✖ | 3.71E-04/✖ | 1.03E+02/✔ | 5.53E-04/✖ |
| | Power Law | 1.55E+03/✔ | 1.49E-04/✖ | 4.51E-03/✖ | 2.18E-03/✖ | 4.13E-04/✖ |

Table 4: Results from *VaCScal* corresponding to Figure 5. Numerical values, along with (✔/✖) and blue/red highlights, indicate the outcomes of Stages I, II, and III of *VaCScal*.

### 6.2.1 Factual Inconsistency Measured via AlignScore

Figure 5 illustrates how factual inconsistency—measured via ALIGNSCORE—scales with FLOPs across LLM families under both power law and exponential models, along with 95% confidence margins. As with the

---

[1] https://github.com/MrYxJ/calculate-flops.pytorch

model size results, similar anomalies persist in FLOP-based scaling—particularly in the E2E dataset (notably with the Mamba family), where inconsistency increases with FLOPs. Similar to the model size-based findings, Appendix I provides a plausible explanation for this aberrant behavior. In Figure 5, E2E also exhibits higher MoE across both scaling models and most LLM families. Outside these exceptions, exponential scaling consistently yields narrower MoEs, indicating more stable fits with lower residual uncertainty. Table 4 summarizes the *VaCScal* evaluation for this FLOP-based analysis. In Stage I, exponential models consistently achieve lower held-out loss. While Stage II includes a few infeasible cases (e.g., E2E and ViGGO with Qwen2.5), exponential scaling is statistically preferred in most model–dataset pairs, as indicated by the frequent blue marks showing significant Vuong test outcomes in Stage III. Here, Qwen2.5 shows conclusions similar to its model size–based scaling behavior for factual inconsistency measured via ALIGNSCORE. Overall, the *VaCScal* results indicate that exponential scaling provides a more robust and reliable model for capturing how factual inconsistency in D2T generation decreases with increasing compute—offering actionable insights for compute-budget planning with fine-tuned LLMs.

**Takeaway.** *Exponential scaling model better captures how factual inconsistency (via ALIGNSCORE) decreases with FLOPs across LLM families, offering a more stable and practical scaling model than power law scaling.*

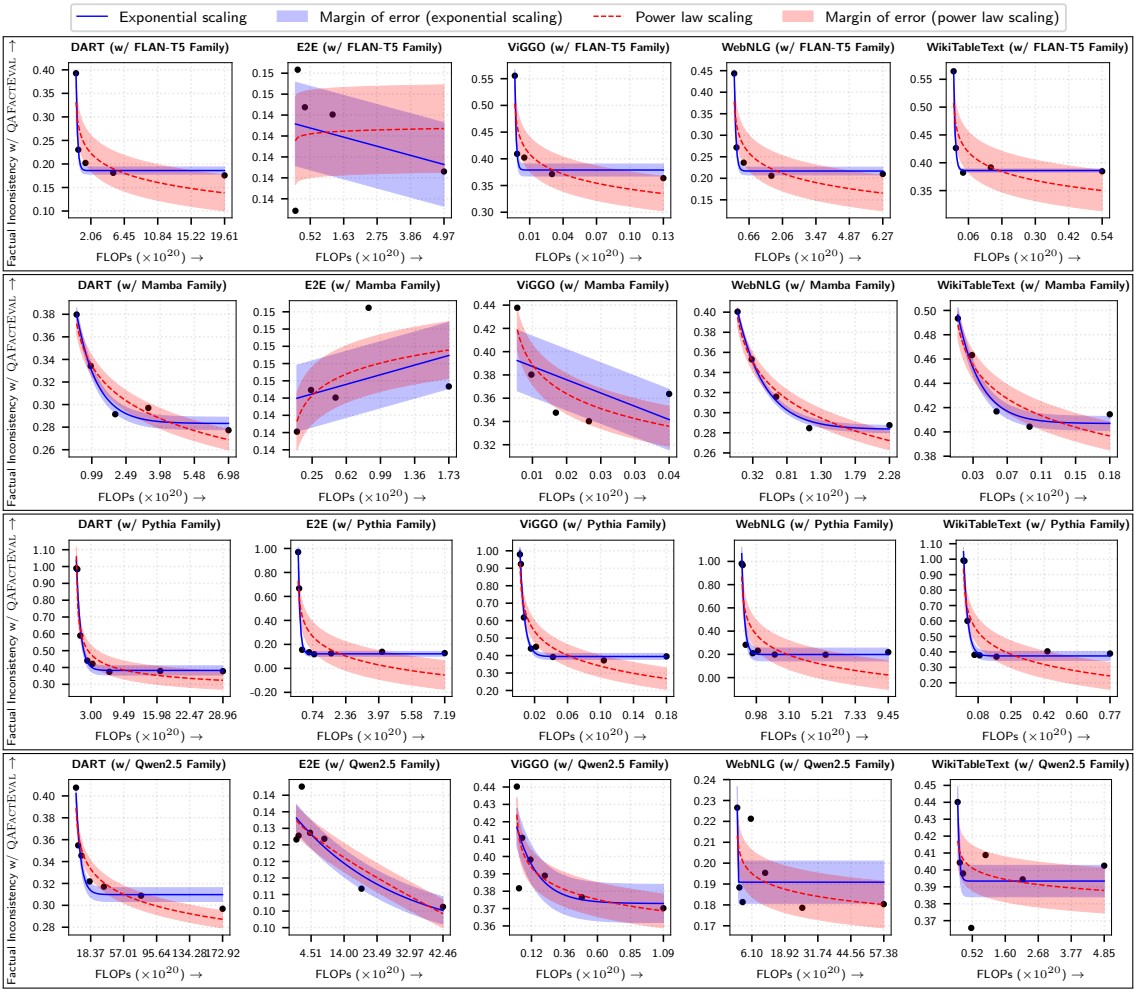

Figure 6: Fitted power law and exponential scaling models (with 95% confidence intervals) for factual inconsistency, measured by QAFACTEVAL, plotted against FLOPs across different LLM families.

### 6.2.2 Factual Inconsistency Measured via QAFactEval

Figure 6 displays fitted power law and exponential scaling curves with 95% confidence intervals (MoE), showing how factual inconsistency—measured using QAFACTEVAL—varies with FLOPs. Again, certain anomalies persist; for instance, factual inconsistency increases with compute for the E2E dataset under the FLAN-T5 and Mamba families. The E2E dataset (within FLAN-T5 and Mamba families) often exhibits large MoEs, indicating high uncertainty. Nevertheless, exponential scaling consistently delivers narrower confidence bounds than power law fits, suggesting more reliable and stable modeling. The associated *VaCScal* outcomes, summarized in Table 5, reinforce this conclusion: most model–dataset pairs show a clear preference for exponential scaling, as indicated by the frequent blue-highlighted Vuong test results, with only a few exceptions (in DART and WebNLG with FLAN-T5). While a few configurations—such as E2E—are deemed infeasible in Stage II, the broader pattern of *VaCScal* results affirms exponential scaling as the superior framework for capturing how factual inconsistency declines with increasing computational effort.

| LLM Family | Scaling | DART | E2E | ViGGO | WebNLG | WikiTableText |
|---|---|---|---|---|---|---|
| FLAN-T5 | Exponential | 2.37E-03/✔ | 6.81E-05/✘ | 4.00E-03/✔ | 2.34E-04/✔ | 1.64E-04/✔ |
| | Power Law | 2.24E+02/✔ | 5.52E-06/✘ | 3.73E-03/✔ | 6.87E-03/✔ | 3.64E-04/✔ |
| Mamba | Exponential | 8.12E-05/✔ | 1.76E-06/✘ | 1.96E-04/✔ | 3.72E-05/✔ | 1.10E-03/✔ |
| | Power Law | 8.31E+08/✘ | 2.60E-04/✘ | 7.56E-04/✘ | 7.46E+05/✘ | 2.66E+21/✘ |
| Pythia | Exponential | 2.81E-03/✔ | 2.32E-02/✔ | 9.32E-04/✔ | 3.67E-01/✔ | 7.35E+01/✔ |
| | Power Law | 5.64E+06/✘ | 5.29E-02/✔ | 9.25E+03/✔ | 1.01E+15/✘ | 4.59E-02/✔ |
| Qwen2.5 | Exponential | 3.80E-04/✔ | 7.23E-05/✔ | 3.41E-04/✘ | 1.91E-04/✘ | 5.13E-04/✘ |
| | Power Law | 6.40E-01/✔ | 1.58E-02/✘ | 1.33E-01/✔ | 1.76E-04/✘ | 4.90E-04/✘ |

Table 5: Results from *VaCScal* corresponding to Figure 6. Numerical values, along with (✔/✘) and blue/red highlights, indicate the outcomes of Stages I, II, and III of *VaCScal*.

**Takeaway.** *Compared to power law scaling, the exponential model more effectively captures the relationship between FLOPs and factual inconsistency (measured via QAFACTEVAL), exhibiting greater stability across LLM families.*

## 7 Discussion

Our findings consistently demonstrate that exponential scaling provides a more accurate and statistically grounded explanation of factual inconsistency in D2T generation, as compared to power law scaling. This conclusion holds across multiple evaluation metrics, scaling factors, and experimental setups. In the main results (section 6), exponential scaling outperformed power law scaling in modeling factual inconsistency, as measured by ALIGNSCORE and QAFACTEVAL, with respect to both model size and compute budget (FLOPs). This trend persists even when alternative factual consistency metrics are used—such as SUMMAC-CONV and UNIEVAL-FACT (Appendix D)—and when evaluated against human annotations (Appendix B), further reinforcing the robustness of the observed scaling behavior. Moreover, exponential scaling remains dominant when other scaling factors are considered, such as the size of the fine-tuning dataset (Appendix G). To ensure that our conclusions are not dependent on a specific fine-tuning or decoding strategy, we conducted additional experiments using Prefix-Tuning (Appendix F) and nucleus sampling (Appendix E). Furthermore, to more comprehensively capture the scaling behavior of factual inconsistency in LLMs for D2T, we also include a full fine-tuning approach, where all trainable model parameters are updated (Appendix H). In all these cases, exponential scaling consistently showed superior performance. Although a few exceptions were observed—particularly with the E2E and ViGGO datasets—these anomalies are limited in scope and do not undermine the broader empirical trend. We hypothesize that this aberrant behavior in E2E and ViGGO stems from two factors: their lower lexical diversity and their closed-domain nature. These characteristics likely make them more prone to overfitting in larger LLMs, leading to the observed inconsistencies. In contrast, the other three D2T datasets, being more diverse and open-domain, mitigate such effects. In Appendix I, we provide a critical analysis of the aberrant behavior exhibited by these two datasets in details. A key enabler of our analysis is the proposed framework *VaCScal*, which offers statistically rigorous validation of scaling models beyond conventional measures such as margin of error (MoE) or held-out loss. While MoE is helpful for visualizing uncertainty, it does not assess the statistical plausibility of a model. In several cases,

models with low MoE failed Stage II of *VaCScal*, due to high *p*-values, indicating that a visually good fit may not always be statistically valid. Furthermore, Stage III of *VaCScal* provides a principled mechanism for comparing competing scaling models, often yielding strong evidence in favor of exponential scaling when both models pass feasibility checks in Stage II. Interestingly, Stage I of *VaCScal* also reveals that exponential scaling tends to achieve lower held-out loss even in settings where neither model passes Stage II feasibility, suggesting its robustness in practice. In summary, our comprehensive evaluation—spanning multiple metrics, datasets, scaling factors, and validation strategies—empirically substantiates the superiority of exponential scaling over power law scaling for capturing factual inconsistency in D2T generation with fine-tuned LLMs.

## 8 Theoretical Implications of Exponential Scaling for Factual Inconsistency in D2T

We aim to provide a mathematical rationale for why factual inconsistency in D2T tasks is expected to follow an exponential scaling pattern with respect to LLM size. We posit that *source-reference divergence* plays a central role in the emergence of factual inconsistency in LLM outputs. Source-reference divergence (Dhingra et al., 2019; Ji et al., 2023) refers to the deviation between the structured input data and the reference text in the training corpus. Such divergence is prevalent in D2T datasets due to heuristic data collection methods, data noise, and the influence of annotators' prior knowledge. Training on these divergent datasets often induces factual inconsistency in model outputs. We further assume that factual inconsistency can be represented through a relative measure of perplexity, which allows us to ground its behavior in a formal scaling law.

**Theoritical Implication 8.1.** *Assume a D2T dataset exhibiting source-reference divergence, and let factual inconsistency be quantified using the relative perplexity of an LLM trained on this dataset. Then,*

$$Factual\ Inconsistency \propto e^{-\xi(N)} \tag{5}$$

*where $\xi(N)$ is a linear or sub-linear function of the model (LLM) size $N$.*

A detailed mathematical interpretation of this theoretical implication is provided in Appendix J.

## 9 Conclusion

This work presents a comprehensive scaling analysis of factual inconsistency in D2T generation with respect to model size, compute (FLOPs), and fine-tuning data size. Contrary to prior studies favoring power law behavior, our experiments across diverse LLM families, datasets, and evaluation metrics consistently reveal that factual inconsistency follows exponential scaling. To rigorously compare scaling models, we introduce *VaCScal*, a three-stage statistical framework that evaluates predictive performance, goodness-of-fit, and model comparison. Exponential scaling outperforms power law across all stages and remains robust across different tuning and decoding strategies. Finally, to reinforce our empirical findings, we provide a mathematical rationale showing that under typical D2T conditions involving source-reference divergence, factual inconsistency is expected to scale exponentially with LLM size. Together, these results establish exponential scaling as a more accurate and theoretically grounded model of factual inconsistency in D2T generation with fine-tuned LLMs.

## Limitation and Future Scope

While our study offers promising insights into the scaling behavior of factual inconsistency in D2T generation, several limitations remain. First, we did not examine the transferability of scaling models across different LLM families or datasets, which could be an important avenue for future work. Second, our analysis is restricted to the D2T task due to computational constraints; extending the framework to other conditional generation tasks such as summarization or dialogue could test its broader applicability. Finally, we focus solely on fine-tuning strategies, leaving out alternative approaches like in-context learning or chain-of-thought prompting—both of which offer interesting directions for future research.

**Broader Impact Statement**

This work highlights that factual inconsistency in LLM-based data-to-text generation exhibits exponential scaling with respect to model size, compute (FLOPs), and fine-tuning data volume. By introducing the *VaC-Scal* framework for statistically grounded model validation and comparison, this study lays the foundation for future efforts in designing more factually consistent and resource-efficient language systems. The accompanying mathematical perspective further deepens understanding of scaling dynamics, offering a principled basis for guiding the development of next-generation LLMs across diverse generation tasks.

**Acknowledgments**

This research is partially supported by the Indo-French Centre for the Promotion of Advanced Research (IFCPAR/CEFIPRA) through CSRP Project No. 6702-2.

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

# A    Experiment Setup

## A.1    D2T Dataset

E2E (Novikova et al., 2017; Dusek et al., 2018) is an MR-to-text dataset from the restaurant domain, containing over 37K instances with an average text length of 21 words. ViGGO (Juraska et al., 2018), another MR-to-text dataset, comprises 7K instances across nine dialogue acts in the video game domain, averaging 14 words per text. Both E2E and ViGGO are closed-domain datasets. WikiTableText (Bao et al., 2018) is an open-domain table-to-text dataset with around 13K pairs derived from Wikipedia tables. DART (Nan et al., 2021) includes nearly 70K knowledge graph triples with an average output length of 34 words. WebNLG (Gardent et al., 2017) is a graph-to-text dataset focused on RDF-to-text generation, containing approximately 38K samples with an average of 30 words per output. Both DART and WebNLG are open-domain datasets. All datasets are sourced from (Kasner et al., 2023) and (Wolf et al., 2020). Key statistics for all datasets are shown in Table 6.

## A.2    LLM Families

We consider six prominent LLM families in our study, spanning three distinct architectural paradigms: decoder-only, encoder-decoder, and state-space model. BLOOM (Scao et al., 2022) is a decoder-only language model trained on the multilingual ROOT dataset. We include five BLOOM variants in our experiments. FLAN-T5 (Chung et al., 2024) is an encoder-decoder model family. Unlike decoder-only models, FLAN-T5

| Dataset | D2T Types | Domain Type | Dataset Size | Source Data | | | Reference | | |
|---------|-----------|-------------|--------------|----------------|---------------|--------------|----------------|---------------|--------------|
| | | | | Average length | Unique Tokens | Total Tokens | Average length | Unique Tokens | Total Tokens |
| E2E | MR-to-text | closed | 36,856 | 27.3 | 125 | 1M | 20.8 | 4.5K | 885K |
| ViGGo | MR-to-text | closed | 6,900 | 29.9 | 618 | 206K | 21.5 | 4.4K | 148K |
| WikiTableText | Table-to-text | open | 13,318 | 35.2 | 29K | 469K | 13.9 | 24K | 185K |
| DART | graph-to-text | open | 70,524 | 34.8 | 44K | 2.5M | 19.3 | 45K | 1.5M |
| WebNLG | graph-to-text | open | 38,872 | 30.4 | 7K | 1.2M | 20.1 | 19K | 905K |

Table 6: Summary of key statistics, D2T types, and domains for the five incorporated datasets. The table presents average length (number of tokens in text), unique tokens, and total tokens for both sources data (linearized to text) and references.

processes input prompts primarily through its encoder. We incorporate five models of varying sizes from this family. OPT (Zhang et al., 2022) consists of decoder-only transformer models ranging from 130M to 13B parameters. OPT family are trained on a mixture of datasets including Reddit, the Pile (Gao et al., 2020), and RoBERTa (Liu et al., 2019) pretraining corpora, following the setup of Brown et al. (2020). We use six OPT models in our experiments. Pythia (Biderman et al., 2023) is a collection of eight decoder-only, GPT-style autoregressive models (70M–12B parameters) employing flash attention. All Pythia models are trained on the Pile dataset using a consistent data ordering scheme. In contrast to these attention-based models, recent work in NLP has explored state-space models as a scalable alternative for modeling long-range dependencies with linear-time inference. To include this model paradigm, we include Mamba (Gu & Dao, 2024), a state-of-the-art state-space language model, and evaluate five different model sizes. Qwen2.5 (Yang et al., 2024) is a decoder-only LLM family built on the Transformer architecture (Vaswani et al., 2017). Our experiments use seven models from this family, covering a wide range of sizes from 0.5B to 72B parameters. The models are pre-trained on an extensive corpus of 18 trillion tokens and further enhanced through post-training techniques, including supervised fine-tuning on over 1 million examples, resulting in strong language understanding and generation capabilities. Qwen2.5 exhibits advanced reasoning skills (e.g., in mathematics and programming) and robust commonsense knowledge. Given its training on structured data such as tables and JSON, we hypothesize that Qwen2.5 can significantly enhance D2T task performance and provide valuable insights into scaling behavior.

### A.3 Fine-tuning and Decoding Strategies

All LLMs are primarily fine-tuned separately on each of the five D2T datasets using two parameter-efficient fine-tuning strategies: QLoRA (Dettmers et al., 2023) (Quantized Low-Rank Adapter) and Prefix-Tuning (Li & Liang, 2021). In the QLoRA setup, we use a reduced rank ($r = 16$), applied primarily to the attention and feedforward modules of the LLMs. For Prefix-Tuning, we use a virtual prefix of 32 tokens. Prefix-Tuning is widely regarded as one of the most effective soft prompt-tuning methods. Prefix-Tuning is applied to all considered LLM families except Mamba, due to unstable initialization issues for virtual tokens in the state-space architecture. For both fine-tuning methods, we fix the learning rate at $1.00e-04$ for trainable parameters. Full fine-tuning (i.e., updating all trainable parameters) is often regarded as essential for accurately capturing scaling behavior. For full fine-tuning , we use a learning rate of $5.00e-05$. Given the critical role of decoding strategies in D2T, we evaluate both deterministic (greedy decoding) and stochastic (nucleus sampling) approaches to provide a comprehensive analysis. For nucleus sampling, we set the nucleus size to 0.95. Most of the results in this paper are based on text generated through greedy decoding.

### A.4 Training Setting and Libraries

In all of our experiments, we use the `SciPy` library (Virtanen et al., 2020) to train both scaling models (power law and exponential) on factual inconsistency results. Along with plotting the scaling models, we include the margin of error (MoE) on residual deviation with a 95% confidence interval, as a measure of the uncertainty in fitting these models. We also use the same `SciPy` library for all hypothesis testing. Vuong's likelihood-ratio test is implemented in Python. For all fine-tuning and model quantization tasks involving the LLM families, we make extensive use of the `transformers` library (Wolf et al., 2020) from HuggingFace. Fine-tuning and evaluation are conducted separately on each D2T dataset, using their respective training

and testing splits. All experiments were conducted using a single NVIDIA A6000 GPU (48 GB) and a single NVIDIA A100 GPU (80 GB). The cumulative computational time across both devices was approximately 22.5 days. Throughout the entire training process, no catastrophic failures or system instabilities were encountered.

### A.5 Input Linearisation and Prompt Structure

Input linearisation and prompt structure are essential components of D2T generation, influencing both language model fine-tuning and inference.

**Input Linearisation.** Linearisation refers to the process of transforming semi-structured input data into a sequential format suitable for LLM-based generation. As discussed earlier, we consider three types of D2T tasks: MR-to-text (E2E and ViGGO), table-to-text (WikiTableText), and graph-to-text (DART and WebNLG). For MR-to-text and table-to-text generation, inputs are linearised as sequences of slot–value pairs in the following format:

```
[(SlotName1, Value1), (SlotName2, Value2), ..., (SlotNameN,
ValueN)]
```

For graph-to-text generation, inputs are linearised as lists of relational triplets in the following format:

```
[(EntityA1, Relation1, EntityB1), (EntityA2, Relation2,
EntityB2), ..., (EntityAN, RelationN, EntityBN)]
```

**Prompt Structure.** Prompt design plays a crucial role in both supervised fine-tuning (and other post-training methods) and inference for D2T tasks. For our experiments, we use the following unified prompt template for both supervised fine-tuning (Zhang et al., 2022; Yang et al., 2024) and inference:

```
Instruction:  Please generate a textual description ('Output')
from the provided structured data ('Input').
Input:  [linearised input data]
Output:  [reference text]
```

# B Scaling Behaviour Based on Model Size of Fine-Tuned (QLoRA) LLM Families With Human Evaluation

Although automatic metrics for evaluating factual consistency have improved in recent years, they still fall short compared to human-based evaluations. To strengthen the validity of exponential scaling for modeling factual inconsistency in D2T, we conducted a human evaluation involving five university student annotators from diverse academic backgrounds. This voluntary task was performed on outputs generated by four LLM families—FLAN-T5, Mamba, OPT, and Qwen2.5. Importantly, this evaluation was limited to QLoRA-fine-tuned versions of these LLM families. Prior to the annotation process, each annotator participated in a brief training session that included five example instances, along with corresponding source data and reference texts. Factual inconsistency was assessed on a Likert scale ranging from 5 (completely inconsistent) to 0 (fully consistent). In addition to assigning scores, annotators were required to justify inconsistencies using one or more of the following categories: (1) *missing fact*, (2) *fabricated fact*, and (3) *irrelevant fact*. A gold-labeled reference set was provided to guide the evaluation process (Table 7). Each annotator independently evaluated 30 generated outputs across five D2T datasets, without consulting others or using external resources. Inter-annotator agreement was high, with a Fleiss' kappa coefficient of 0.79, indicating strong reliability.

| source | reference | generated text | label (Likert Scale 0-5, with reasons) |
|---|---|---|---|
| ('MotorSport Vision', 'city', 'Fawkham') | MotorSport Vision is located in Fawkham . | Fawkham is in the UK. | 5 (inconsistent: irrelevant fact) |
| (('subjtitle', 'kf tirana'), ('subjsubtitle', 'presidents history'), ('name', 'lutfi nuri'), ('from–to', '1998–99')) | lutfi nuri was president of kf tirana during 1998–99 . | lutfi nuri president of kf tirana who took office 1998–99 . | 0 (consistent) |
| (('subjtitle', 'aachen'), ('subjsubtitle', 'demographics'), ('year', '2015'), ('population', '245,885')) | the population of aachen 245,885 in 2015 . | 245,885 people in aachen lived in 2015 . | 0 (consistent) |
| (('subjtitle', 'fc rubin kazan'), ('subjsubtitle', 'average attendance'), ('year', '2011'), ('reg . season', '16,380')) | the average attendance of fc rubin kazanin 2011 were 16,380 . | fc rubin kazan got 16,380 attendences in 2011 . | 1 (almost consistent: missing 'average') |
| (('name', 'Clowns'), ('eatType', 'pub'), ('near', 'The Sorrento')) | Close to The Sorrento you can find Clowns pub. | Clowns pub serves food for under £20 and is located near The Sorrento. | 3 (inconsistent: fabricated fact) |

Table 7: The five short sample given to each annotators for training before annotation task.

Figure 7 presents the fitted power law and exponential scaling curves, along with 95% confidence intervals (MoE), based on human-annotated assessments of factual inconsistency across varying model sizes (in terms of parameter count). Across all D2T datasets, exponential scaling consistently exhibits lower MoEs than power law scaling, indicating greater stability and confidence under human evaluation. The corresponding *VaCScal* results are shown in Table 8, where, in almost all datasets (except E2E), the LLM families favor exponential scaling. This is evidenced by the blue-highlighted cells denoting strong statistical preference for exponential scaling in Stage III of the Vuong test. While some exceptions are observed in the E2E dataset, even in these cases, exponential scaling achieves lower held-out error in Stage I compared to power law scaling—indicating superior predictive performance. Although in two rare cases, where power law scaling outperforms exponential scaling for the DART and WebNLG datasets with FLAN-T5 models, exponential scaling remains competitive in Stages I and II of the *VaCScal* framework. Except for these rare cases, all other consistent *VaCScal* outcomes further support the suitability of exponential scaling for modeling the relationship between model size and factual inconsistency in D2T generation.

**Takeaway.** *In human evaluations across the FLAN-T5, Mamba, OPT, and Qwen2.5 LLM families, exponential scaling shows stronger acceptance in capturing the decrease of factual inconsistency with increasing model size, offering a more stable and practically reliable model than power law scaling.*

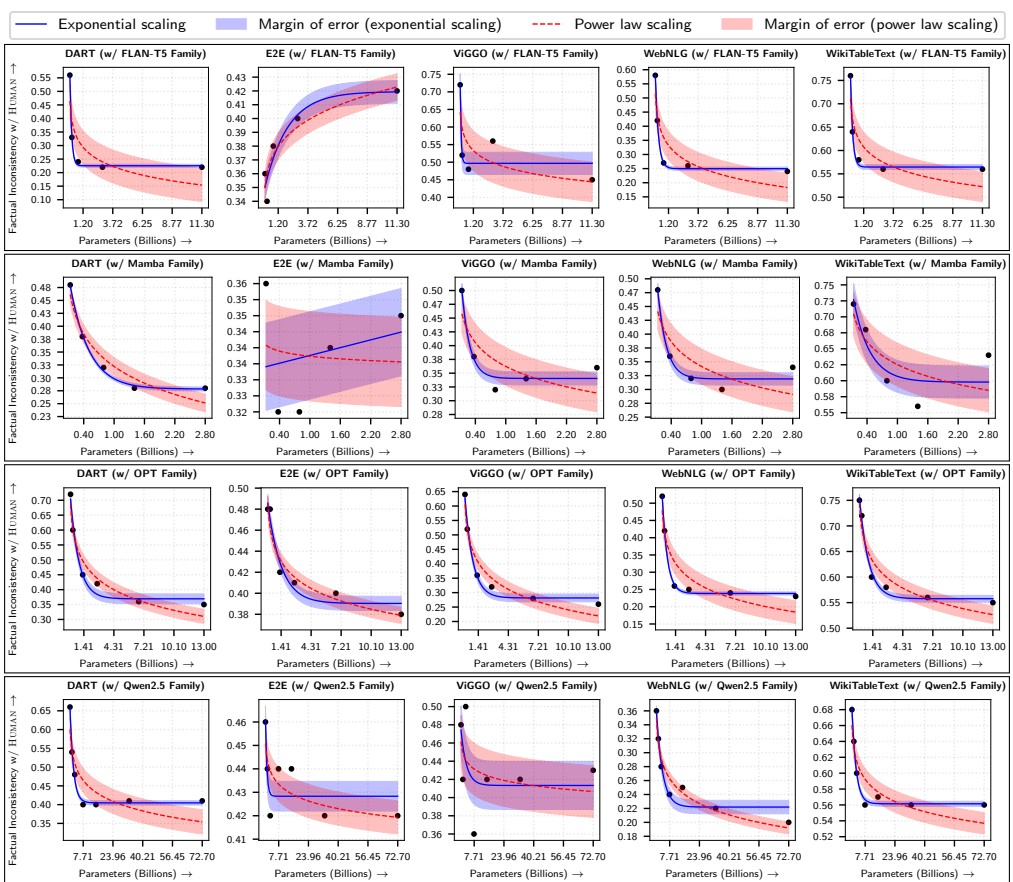

Figure 7: Fitted power law and exponential scaling models (with 95% confidence intervals) for factual inconsistency in D2T, based on HUMAN evaluation, plotted against model size for QLoRA fine-tuned FLAN-T5, Mamba, OPT, and Qwen2.5 LLM families.

| LLM Family | Scaling | DART | E2E | ViGGO | WebNLG | WikiTableText |
|---|---|---|---|---|---|---|
| FLAN-T5 | Exponential | 1.14E-04/✔ | 4.31E-05/✘ | 1.54E-03/✔ | 2.48E-03/✔ | 4.35E+00/✔ |
| | Power Law | 1.54E-04/✔ | 7.06E-04/✘ | 8.62E-05/✔ | 5.05E-04/✔ | 6.02E-04/✔ |
| Mamba | Exponential | 3.28E-06/✔ | 8.17E-05/✘ | 1.69E-04/✔ | 2.84E-04/✔ | 6.68E-04/✘ |
| | Power Law | 4.51E+05/✘ | 1.47E-03/✘ | 2.53E+176/✘ | 5.66E-01/✘ | 8.87E+13/✘ |
| OPT | Exponential | 1.25E-03/✔ | 4.29E-04/✔ | 4.13E-03/✔ | 2.54E-04/✔ | 2.84E-03/✔ |
| | Power Law | 7.00E+01/✘ | 1.26E+00/✘ | 6.51E+02/✘ | 1.58E+10/✘/◗ | 2.03E-03/✘ |
| Qwen2.5 | Exponential | 4.51E-04/✔ | 1.26E-04/✘ | 4.05E+00/✔ | 6.08E-05/✔ | 7.98E-05/✔ |
| | Power Law | 3.88E+00/✔ | 6.44E-04/✘ | 1.03E-04/✘ | 9.36E-05/✘ | 4.05E-01/✘ |

Table 8: Results from *VaCScal* corresponding to Figure 7. Numerical values, along with (✔/✘) and blue/red highlights, indicate the outcomes of Stages I, II, and III of *VaCScal*.

# C Result: Scaling Behavior Based on Model Size and FLOPs for Fine-Tuned (QLoRA) BLOOM and OPT Families

Here, we present the scaling behavior of fine-tuned (QLoRA) BLOOM and OPT families with respect to factual inconsistency, measured using AlignScore and QAFactEval. Figure 8 and Figure 9 illustrate the fitted power law and exponential scaling models for both families, based on model size and FLOPs, respectively. We find that scaling trends based on model size and FLOPs are largely consistent with each other. Similar to the other four LLM families, exponential scaling generally provides a better fit than power law scaling, particularly for the OPT family, as indicated by narrower margins of error (MoE). Although BLOOM exhibits relatively higher MoEs, exponential scaling still consistently achieves lower MoEs compared to power law scaling. Despite the aberrant scaling behavior observed for the E2E and ViGGO datasets (Appendix I), the corresponding *VaCScal* results in Table 9 and Table 10 strongly support the superiority of exponential scaling over power law scaling across all three validation stages.

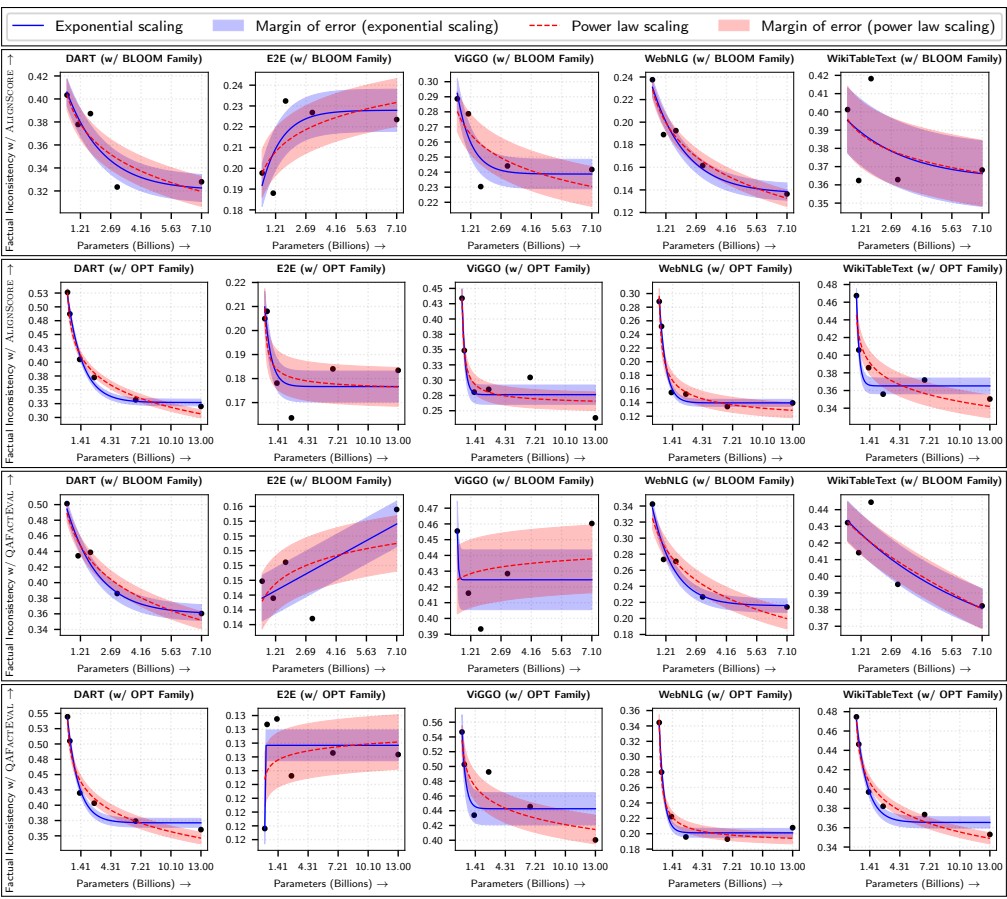

Figure 8: Fitted power law and exponential scaling models (with 95% confidence intervals) for factual inconsistency, measured by QAFactEval, plotted against model size across different LLM families.

**Takeaway:** *In D2T, exponential scaling demontrate the factual inconsistency (measured by AlignScore and QAFactEval) with increasing model size and FLOPs more reliably than power law scaling across BLOOM and OPT families, except a few minor exception.*

| Metric | LLM family | Scaling | DART | E2E | ViGGO | WebNLG | WikiTableText |
|---|---|---|---|---|---|---|---|
| AlignScore | BLOOM | Exponential | 1.84E-04/✗ | 6.29E+03/✗ | 5.82E+03/✗ | 2.51E-04/✔ | 1.43E+03/✗ |
| | | Power Law | 6.92E-04/✗ | 6.65E-04/✗ | 3.30E-04/✗ | 2.43E-02/✗ | 1.55E-03/✗ |
| | OPT | Exponential | 1.64E-04/✔ | 1.15E-01/✗ | 1.37E-03/✔ | 3.66E-04/✔ | 9.04E-04/✔ |
| | | Power Law | 2.96E-03/✗ | 1.16E+82/✗ | 3.72E+81/✗ | 1.94E-03/✗ | 1.10E+03/✔ |
| QAFactEval | BLOOM | Exponential | 2.07E-04/✔ | 7.81E-05/✗ | 7.41E-04/✗ | 4.81E-04/✔ | 3.76E-04/✗ |
| | | Power Law | 1.00E-01/✗ | 2.45E-03/✗ | 8.86E-04/✗ | 3.10E-02/✗ | 4.43E+06/✗ |
| | OPT | Exponential | 1.13E-04/✔ | 5.28E-05/✗ | 1.28E-03/✗ | 3.63E-03/✗ | 1.35E-04/✔ |
| | | Power Law | 2.00E+01/✗ | 9.51E-02/✗ | 1.63E-02/✗ | 5.69E+02/✔ | 4.42E+00/✗ |

Table 9: Results from *VaCScal* corresponding to Figure 8. Numerical values, along with (✔/✗) and blue/red highlights, indicate the outcomes of Stages I, II, and III of *VaCScal*.

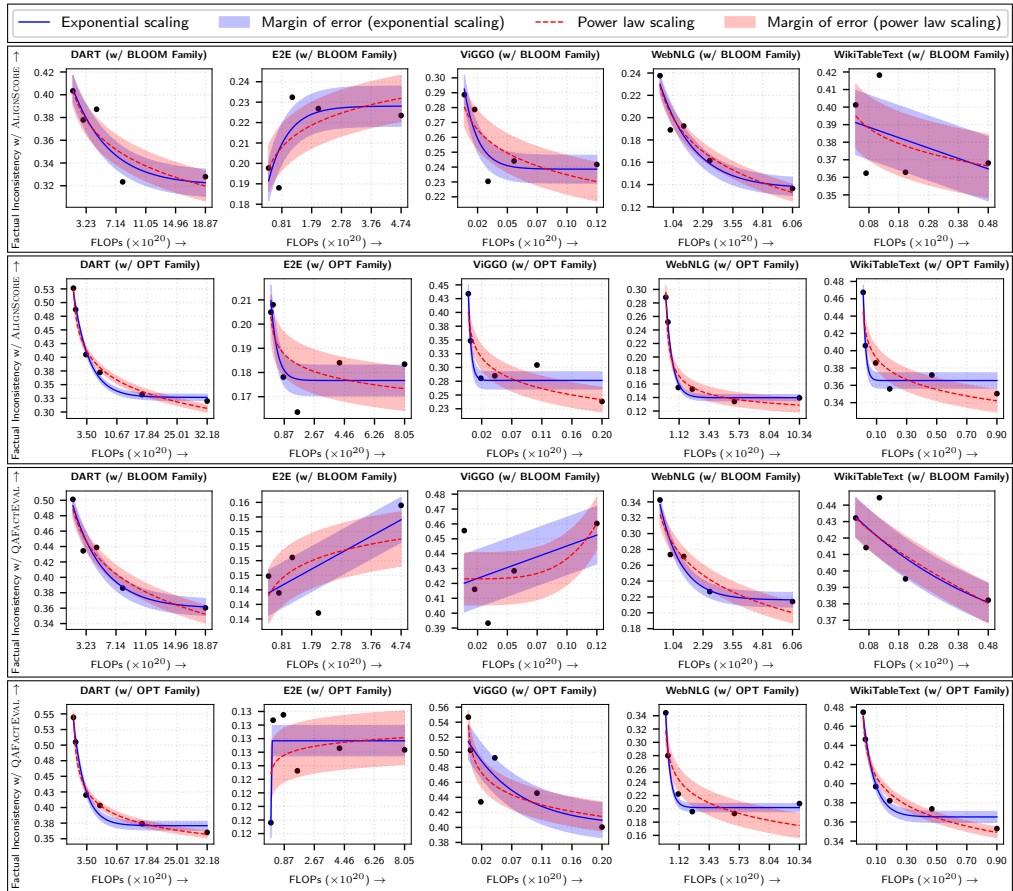

Figure 9: Fitted power law and exponential scaling models (with 95% confidence intervals) for factual inconsistency, measured by AlignScore, plotted against FLOPs across different LLM families.

| Metrics | LLM Family | Scaling | DART | E2E | ViGGO | WebNLG | WikiTableText |
|---|---|---|---|---|---|---|---|
| AlignScore | BLOOM | Exponential | 4.77E-03/✗ | 1.35E+03/✗ | 1.13E+03/✗ | 3.60E-04/✔ | 2.22E+02/✗ |
| | | Power Law | 1.64E-01/✗ | 1.39E-02/✗ | 2.83E+198/✗ | 2.64E-04/✗ | 1.35E-03/✗ |
| | OPT | Exponential | 3.40E-04/✔ | 1.83E-04/✗ | 1.19E-03/✔ | 1.16E-04/✔ | 4.44E+02/✔ |
| | | Power Law | 9.22E+01/✗ | 1.71E-04/✗ | 5.96E-04/✗ | 1.69E+08/✗ | 6.29E+02/✔ |
| QAFactEval | BLOOM | Exponential | 4.31E-04/✔ | 1.33E-04/✗ | 6.10E-04/✗ | 6.60E-04/✔ | 2.27E-04/✗ |
| | | Power Law | 4.75E-02/✗ | 6.22E-05/✗ | 4.54E-04/✗ | 2.88E-02/✗ | 1.54E-02/✗ |
| | OPT | Exponential | 1.85E-04/✔ | 2.00E-05/✗ | 1.03E-03/✗ | 2.97E-03/✔ | 1.23E-04/✔ |
| | | Power Law | 2.59E+01/✗ | 1.71E-04/✗ | 6.37E+04/✗ | 1.30E+07/✔ | 1.52E+00/✗ |

Table 10: Results from *VaCScal* corresponding to Figure 9. Numerical values, along with (✔/✗) and blue/red highlights, indicate the outcomes of Stages I, II, and III of *VaCScal*.

# D    Result: Scaling Behavior Based on Model Size of Fine-Tuned (QLoRA) LLM Families

## D.1    Factual Inconsistency Measured via SummaC-conv

Figure 10 presents fitted power law and exponential scaling curves (with 95% confidence intervals) for factual inconsistency measured via SUMMAC-CONV, plotted against model size across all six LLM families. Although some anomalies appear—particularly in the E2E dataset with BLOOM—exponential scaling consistently yields lower MoE and lower held-out loss. Table 11 shows corresponding *VaCScal* results, where exponential models pass statistical tests more frequently and are preferred by Vuong's test in most settings. These *VaCScal* results support the robustness of exponential scaling in modeling factual inconsistency—measured via SUMMAC-CONV—with respect to LLM model size across families.

**Takeaway:**    *Exponential scaling captures better the decline of factual inconsistency (measured by SUMMAC-CONV) with increasing model size more reliably than power law scaling across LLM families, supported by tighter confidence margins and stronger statistical validation from VaCScal.*

| LLM Family | Scaling | DART | E2E | ViGGO | WebNLG | WikiTableText |
|---|---|---|---|---|---|---|
| BLOOM | Exponential | 1.79E-04/✘ | 2.39E+02/✘ | 2.45E-04/✘ | 5.40E-04/✘ | 2.20E-05/✔ |
| | Power Law | 2.72E-02/✘ | 2.98E-02/✘ | 2.11E-04/✘ | 7.86E-03/✘ | 9.19E-03/✘ |
| FLAN-T5 | Exponential | 6.46E-04/✔ | 9.07E-06/✘ | 2.39E-04/✔ | 2.83E-04/✔ | 3.13E+00/✔ |
| | Power Law | 2.12E-03/✔ | 1.35E-05/✘ | 2.43E+01/✘ | 2.96E-04/✔ | 4.74E-05/✔ |
| Mamba | Exponential | 7.88E-05/✔ | 1.76E-06/✘ | 1.88E+01/✘ | 1.52E-04/✔ | 3.04E-04/✔ |
| | Power Law | 8.26E-06/✘ | 1.00E-04/✘ | 1.08E+168/✘ | 1.14E-04/✘ | 5.13E+11/✘ |
| OPT | Exponential | 1.92E-04/✔ | 2.85E-01/✘ | 6.21E-04/✘ | 3.48E-05/✔ | 4.75E-04/✔ |
| | Power Law | 3.94E-01/✘ | 8.17E+80/✘ | 1.42E-04/✘ | 6.67E+10/✘ | 5.98E+04/✔ |
| Pythia | Exponential | 1.38E-03/✔ | 1.05E-03/✔ | 8.16E-04/✔ | 5.51E-03/✔ | 4.07E-04/✔ |
| | Power Law | 5.21E-03/✘ | 1.25E-02/✔ | 3.14E-03/✘ | 1.42E+09/✘ | 7.34E-03/✘ |
| Qwen2.5 | Exponential | 4.96E-04/✔ | 1.15E-04/✔ | 1.21E-03/✘ | 2.12E-04/✔ | 2.52E-04/✘ |
| | Power Law | 1.38E-03/✔ | 3.70E-03/✘ | 3.08E-04/✘ | 1.57E-02/✘ | 7.73E-03/✘ |

Table 11: Results from *VaCScal* corresponding to Figure 10. Numerical values, along with (✔/✘) and blue/red highlights, indicate the outcomes of Stages I, II, and III of *VaCScal*.

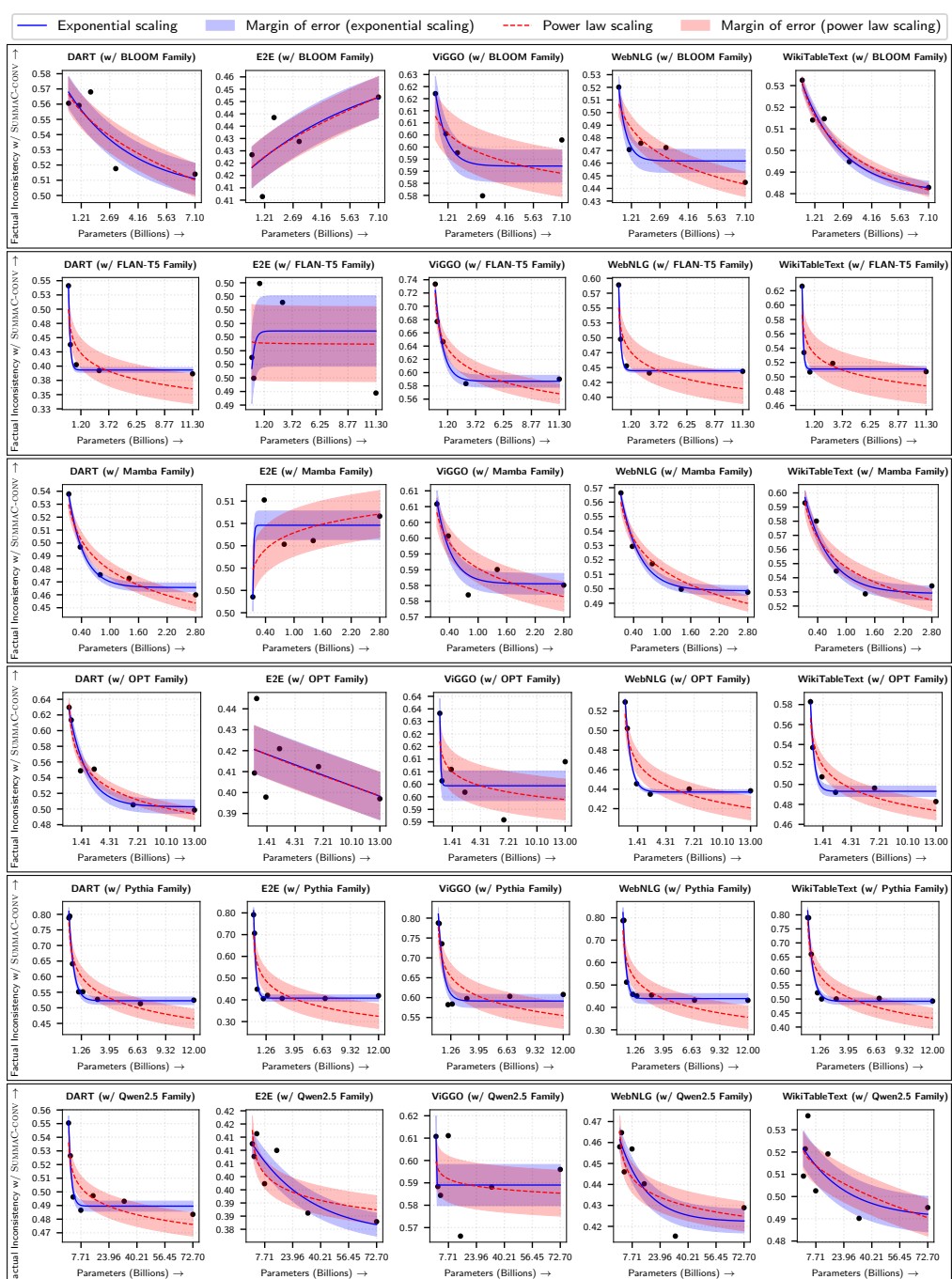

Figure 10: Fitted power law and exponential scaling models (with 95% confidence intervals) for factual inconsistency, measured by SUMMAC-CONV, plotted against model size across different LLM families.

## D.2 Factual Inconsistency Measured via UniEval-fact

Figure 11 shows the fitted power law and exponential scaling curves with 95% confidence intervals (MoE) for factual inconsistency measured by UNIEVAL-FACT. As with other metrics (ALIGNSCORE, QAFACTEVAL, SUMMAC-CONV), anomalies persist—most notably in the E2E dataset and BLOOM family, which exhibit increasing inconsistency or large MoEs. Despite these outliers, exponential scaling consistently yields substantially lower MoEs across all cases compared to power law scaling, indicating reduced uncertainty in

estimation. The corresponding *VaCScal* results (Table 12) further support this observation: exponential scaling achieves significantly lower held-out loss in Stage I. In Stages II and III, exponential models also demonstrate greater statistical feasibility (marked with ✔) and receive more support from Vuong's test (highlighted in blue) than their power law counterparts. Overall, *VaCScal* results affirm that exponential scaling more effectively captures how factual inconsistency varies with model size in D2T generation under UNIEVAL-FACT.

**Takeaway.** *Exponential scaling more reliably models the relationship between model size of LLMs and factual inconsistency in D2T tasks, as measured by UNIEVAL-FACT, outperforming power law scaling despite a few dataset-specific anomalies.*

| LLM Family | Scaling | DART | E2E | ViGGO | WebNLG | WikiTableText |
|---|---|---|---|---|---|---|
| BLOOM | Exponential | 3.07E-04/✔ | 1.39E-04/✘ | 6.31E-04/✘ | 4.78E-06/✔ | 3.45E-04/✔ |
| | Power Law | 3.25E-02/✘ | 8.95E-03/✘ | 1.49E-04/✘ | 1.06E-02/✘ | 1.36E-04/✘ |
| FLAN-T5 | Exponential | 2.15E-04/✔ | 5.01E-05/✘ | 8.94E-04/✔ | 1.26E-03/✔ | 7.04E-01/✔ |
| | Power Law | 2.64E-04/✔ | 4.26E+00/✘ | 1.74E-03/✘ | 1.74E-03/✔ | 8.80E-05/✔ |
| Mamba | Exponential | 4.89E-05/✔ | 1.22E-04/✘ | 1.98E+02/✔ | 4.13E-05/✔ | 2.92E-04/✔ |
| | Power Law | 3.56E+06/✘ | 1.18E-04/✘ | 4.22E+170/✘ | 3.91E-04/✘ | 1.09E-03/✘ |
| OPT | Exponential | 1.13E-03/✔ | 8.96E-05/✘ | 2.81E-03/✔ | 5.28E-01/✔ | 1.23E-04/✔ |
| | Power Law | 3.08E+02/✘ | 5.89E-02/✘ | 6.76E-04/✔ | 1.20E+87/✘ | 3.69E+03/✔ |
| Pythia | Exponential | 1.06E-02/✔ | 1.06E-02/✔ | 7.36E-04/✔ | 6.82E-03/✔ | 4.14E-02/✔ |
| | Power Law | 1.36E+05/✘ | 2.06E+13/✔ | 2.11E+03/✘ | 4.45E+10/✘ | 4.23E+00/✘ |
| Qwen2.5 | Exponential | 4.51E-04/✔ | 1.26E-04/✘ | 4.05E+00/✔ | 6.08E-05/✔ | 7.98E-05/✔ |
| | Power Law | 3.88E+00/✔ | 6.44E-04/✘ | 1.03E-04/✘ | 9.36E-05/✘ | 4.05E-01/✘ |

Table 12: Results from *VaCScal* corresponding to Figure 11. Numerical values, along with (✔/✘) and blue/red highlights, indicate the outcomes of Stages I, II, and III of *VaCScal*.

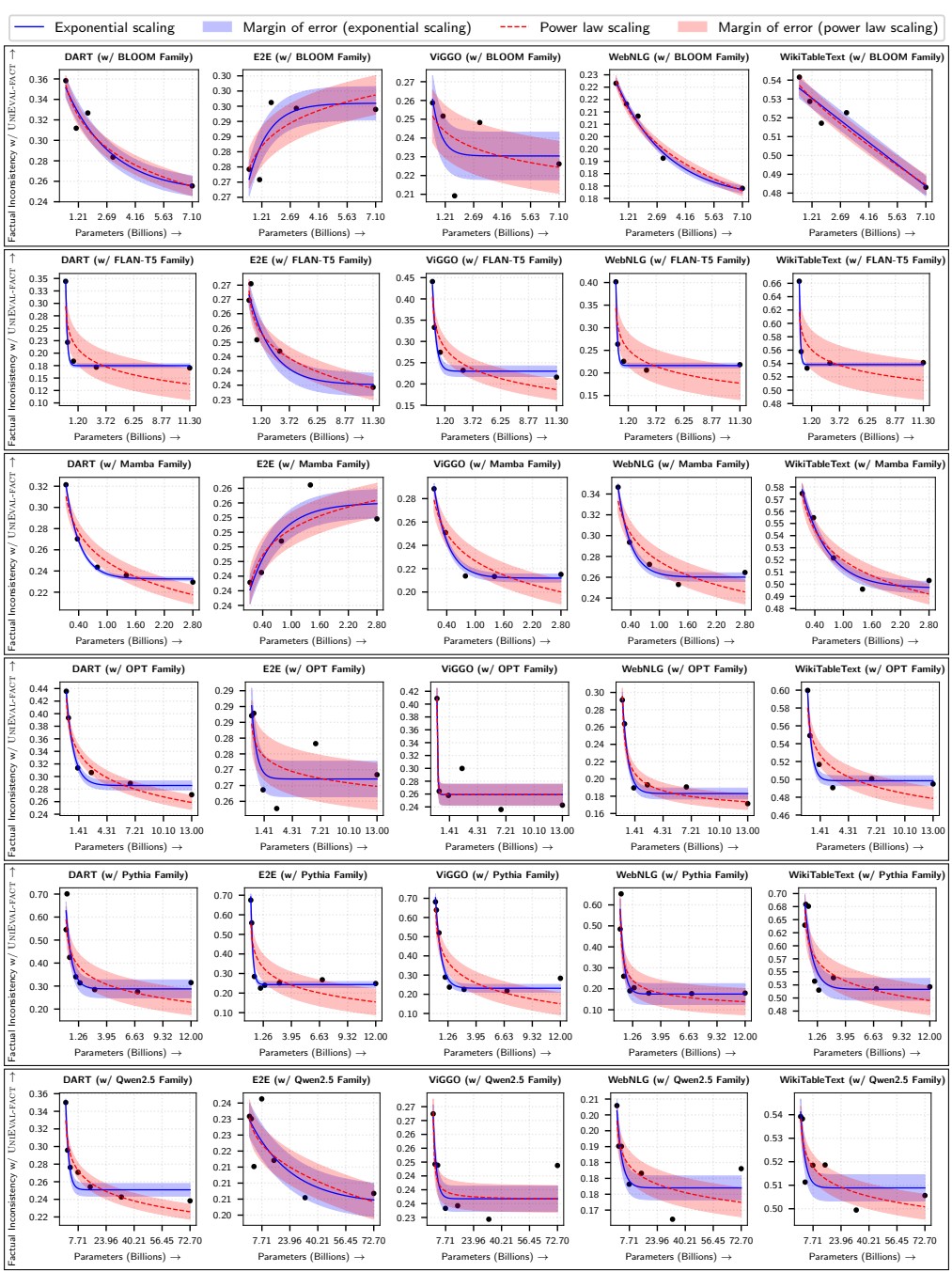

Figure 11: Fitted power law and exponential scaling models (with 95% confidence intervals) for factual inconsistency, measured by UNIEVAL-FACT, plotted against model size across different LLM families.

# E Results: Scaling Behavior Based on Model Size of Fine-Tuned (QLoRA) LLM Families with Nucleus Sampling

As discussed in section 7, all prior results were based on greedy decoding. However, decoding strategy significantly influences the factual inconsistency of LLM outputs. To evaluate the robustness of our findings favoring exponential scaling over power law scaling, we conduct additional experiments using nucleus sampling (with a nucleus probability of 0.95)—a widely used stochastic decoding method. For this setup, five LLM families (BLOOM, FLAN-T5, Mamba, OPT, and Pythia) are fine-tuned on the five D2T datasets using QLoRA, and factual inconsistency is assessed using ALIGNSCORE and QAFACTEVAL.

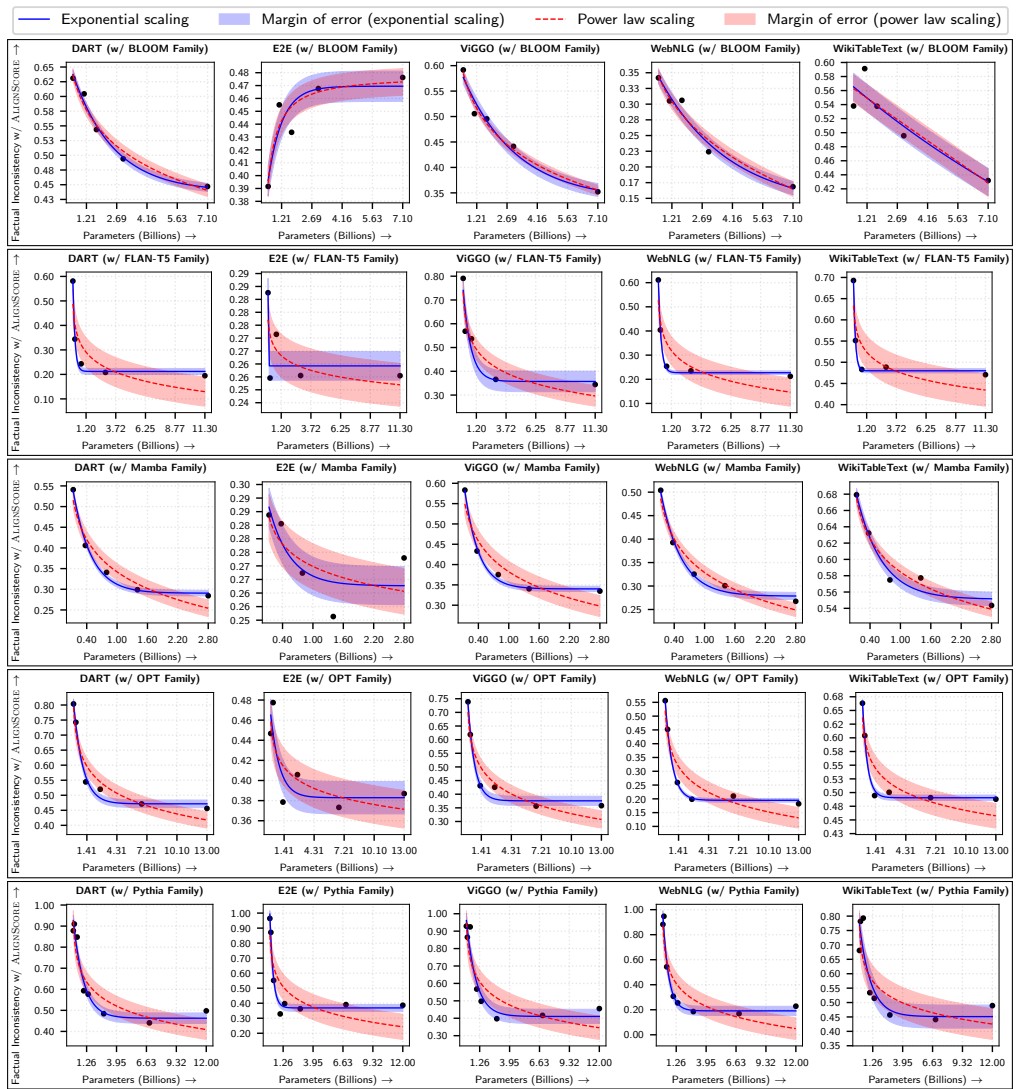

Figure 12: Fitted power law and exponential scaling models (with 95% confidence intervals) for factual inconsistency, measured by ALIGNSCORE, plotted against model size across different LLM families. Here we have considered nucleus sampling for text generation.

Figure 12 and Figure 13 show fitted power law and exponential scaling curves (with 95% confidence intervals) for factual inconsistency—measured by ALIGNSCORE and QAFACTEVAL—under nucleus sampling ($p = 0.95$). The trends largely mirror those from greedy decoding: exponential scaling yields consistently lower MoEs, indicating greater stability. Exceptions such as the E2E dataset and the BLOOM family persist, with large uncertainties and irregular patterns. *VaCScal* results in Table 13 and Table 14 reaffirm exponential

| LLM Family | Scaling | DART | E2E | ViGGO | WebNLG | WikiTableText |
|---|---|---|---|---|---|---|
| BLOOM | Exponential | 2.99E-04/✔ | 1.43E-03/✘ | 1.62E-03/✔ | 2.60E-04/✔ | 2.03E-03/✘ |
| | Power Law | 2.63E+02/✘ | 2.82E-04/✘ | 3.84E-01/✘ | 2.68E-03/✘ | 5.65E-01/✘ |
| FLAN-T5 | Exponential | 2.59E-04/✔ | 5.44E-04/✘ | 7.12E-03/✔ | 2.59E-04/✔ | 9.13E-05/✔ |
| | Power Law | 2.26E-03/✔ | 5.46E-06/✘ | 2.55E-01/✘ | 4.34E+08/✘ | 5.38E+14/✔ |
| Mamba | Exponential | 1.01E-03/✔ | 5.39E-05/✘ | 1.39E-03/✔ | 4.22E-04/✔ | 1.97E-04/✔ |
| | Power Law | 3.88E+03/✘ | 1.05E-03/✘ | 6.78E+05/✘ | 2.51E+02/✘ | 5.08E+03/✘ |
| OPT | Exponential | 9.81E-04/✔ | 8.72E-01/✘ | 3.64E-04/✔ | 2.68E-04/✔ | 3.99E-03/✔ |
| | Power Law | 9.50E+03/✘ | 1.86E+79/✘ | 2.37E+03/✘ | 7.56E-03/✘ | 1.51E-03/✘ |
| Pythia | Exponential | 5.34E-03/✔ | 3.06E-01/✔ | 3.18E-03/✔ | 5.82E-03/✔ | 4.26E-03/✔ |
| | Power Law | 1.56E-02/✘ | 1.97E-02/✔ | 2.17E+03/✘ | 8.13E+04/✘ | 1.24E+04/✘ |

Table 13: Results from *VaCScal* corresponding to Figure 12. Numerical values, along with (✔/✘) and blue/red highlights, indicate the outcomes of Stages I, II, and III of *VaCScal*.

scaling as the preferred model across most dataset–model combinations. These findings confirm that our conclusions hold under stochastic decoding, consistent with the results obtained using greedy decoding across all our experiments.

**Takeaway:** *Exponential scaling remains the more robust and statistically valid model for capturing factual inconsistency in D2T generation—even under stochastic decoding with nucleus sampling—further validating the generalizability of our findings across decoding strategies.*

| LLM Family | Scaling | DART | E2E | ViGGO | WebNLG | WikiTableText |
|---|---|---|---|---|---|---|
| BLOOM | Exponential | 5.01E-03/✔ | 7.46E+01/✘ | 4.92E-03/✘ | 2.76E-03/✔ | 1.49E-03/✘ |
| | Power Law | 8.35E+00/✘ | 5.92E-04/✘ | 9.12E-03/✘ | 2.51E-03/✘ | 2.84E+05/✘ |
| FLAN-T5 | Exponential | 2.80E-04/✔ | 6.21E-05/✘ | 8.37E-03/✘ | 3.01E-04/✔ | 3.11E-03/✔ |
| | Power Law | 5.65E+05/✔ | 1.12E-03/✘ | 8.98E-02/✘ | 5.46E-03/✘ | 1.81E+07/✔ |
| Mamba | Exponential | 3.32E-04/✔ | 5.82E-05/✔ | 9.18E+01/✔ | 1.85E-04/✔ | 3.01E-04/✔ |
| | Power Law | 2.23E+03/✘ | 2.81E-06/✔ | 7.28E-04/✘ | 1.74E+01/✘ | 1.37E-04/✘ |
| OPT | Exponential | 2.83E-03/✔ | 1.58E-05/✔ | 2.14E-03/✔ | 1.13E-03/✔ | 2.64E-02/✔ |
| | Power Law | 2.31E+01/✘ | 6.34E-01/✘ | 1.02E+08/✘ | 3.28E+05/✘ | 1.24E+08/✘ |
| Pythia | Exponential | 1.30E-03/✔ | 2.67E-03/✔ | 3.60E-03/✔ | 7.31E-03/✔ | 5.69E-04/✔ |
| | Power Law | 4.16E+08/✘ | 4.99E+07/✔ | 2.04E+02/✘ | 2.90E-02/✘ | 4.00E+10/✘ |

Table 14: Results from *VaCScal* corresponding to Figure 13. Numerical values, along with (✔/✘) and blue/red highlights, indicate the outcomes of Stages I, II, and III of *VaCScal*.

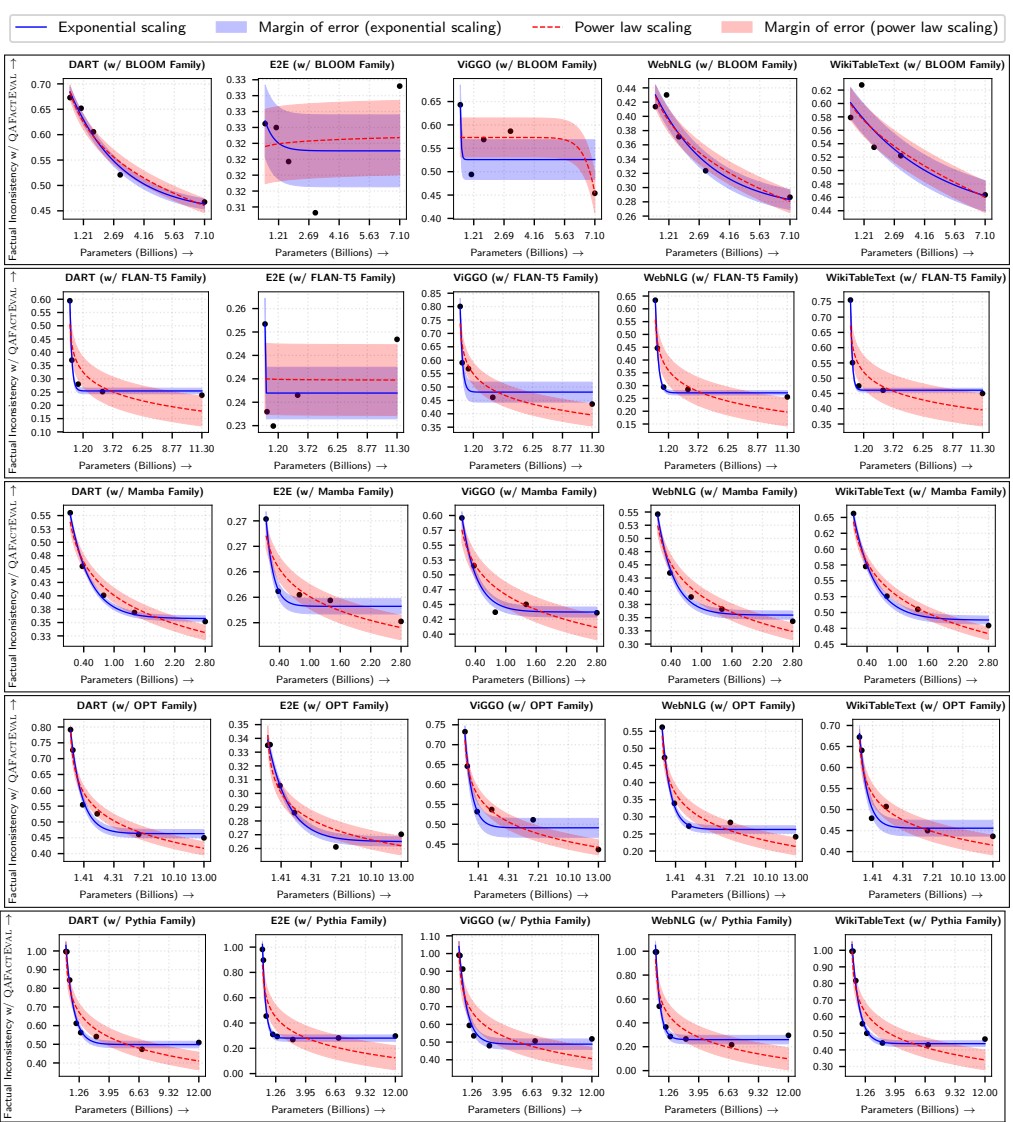

Figure 13: Fitted power law and exponential scaling models (with 95% confidence intervals) for factual inconsistency, measured by QAFACTEVAL, plotted against model size across different LLM families. Here we have considered nucleus sampling for text generation.

# F    Result: Scaling Behavior Based on Model Size of Fine-Tuned (Prefix-Tuning) LLM Families

In this section, we report results from experiments where LLM families (BLOOM, FLAN-T5, OPT, and Pythia) were fine-tuned using Prefix-Tuning, to verify the robustness of our conclusions across different fine-tuning strategies. We compare these results against the previously used QLoRA setup. Factual inconsistency is evaluated using ALIGNSCORE and QAFACTEVAL, as these metrics were sufficient to generalize the trends under QLoRA. Note that Prefix-Tuning was not applied to the Mamba family due to stability issues encountered during fine-tuning.

## F.1    Factual Inconsistency Measured via AlignScore and QAFactEval.

Here we present experimental results using *Prefix-Tuning* as an alternative fine-tuning strategy to the previously employed QLoRA. This comparison aims to assess the robustness of our findings across different fine-tuning approaches. We evaluate factual inconsistency using two representative automatic metrics— ALIGNSCORE and QAFACTEVAL—which have consistently captured key trends and are sufficient to support our conclusions. Note that the Mamba family was excluded from this setup due to instability encountered during Prefix-Tuning.

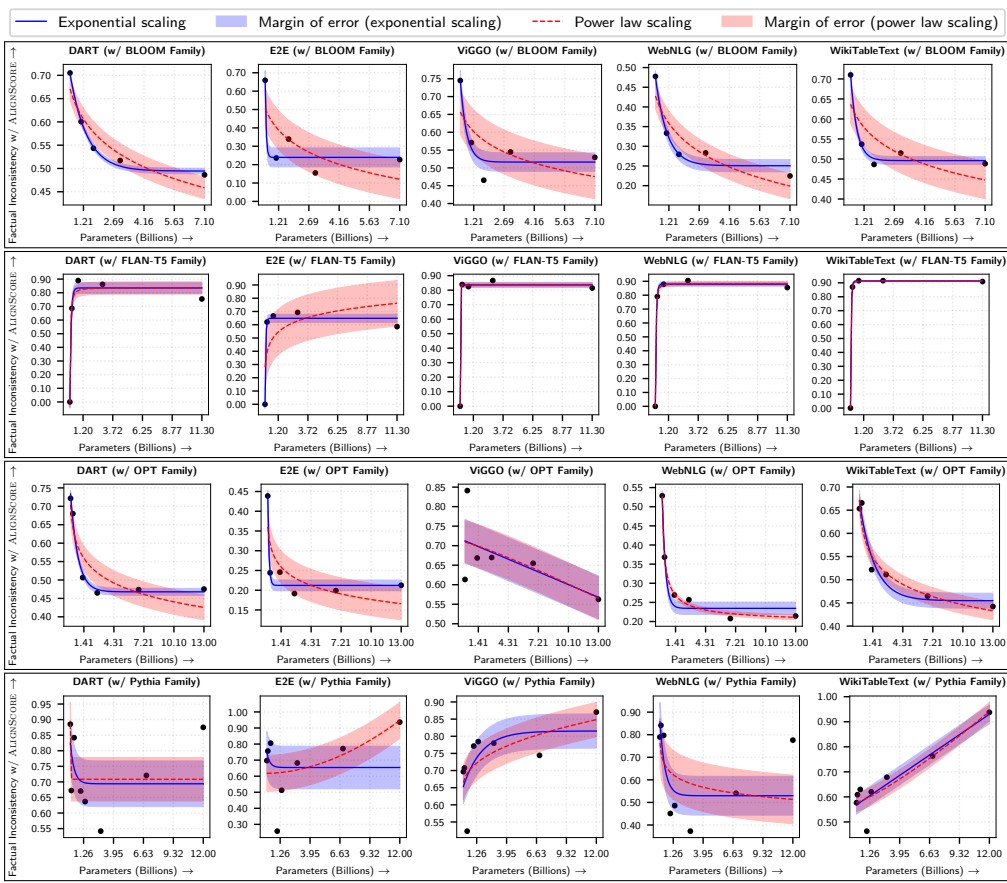

Figure 14: Fitted power law and exponential scaling models (with 95% confidence intervals) for factual inconsistency, measured by ALIGNSCORE, plotted against model size across different LLM families. Here we consider Prefix-Tuning for fine-tuning all LLM families.

Figure 14 and Figure 15 present the fitted power law and exponential scaling curves with 95% confidence intervals (MoE) for factual inconsistency, measured using ALIGNSCORE and QAFACTEVAL, plotted against model size (in terms of parameter count). For ALIGNSCORE, both the FLAN-T5 and Pythia families exhibit

| LLM Family | Scaling | DART | E2E | ViGGO | WebNLG | WikiTableText |
|---|---|---|---|---|---|---|
| BLOOM | Exponential | 2.14E-04/✔ | 1.16E-02/✘ | 4.73E+03/✘ | 2.78E-05/✔ | 3.00E+03/✔ |
| | Power Law | 1.31E+05/✘ | 6.16E-03/✘ | 3.25E-03/✘ | 1.08E-03/✘ | 1.17E+203/✔ |
| FLAN-T5 | Exponential | 1.09E-02/✘ | 3.01E-02/✘ | 5.14E-03/✘ | 3.88E-02/✘ | 3.08E-03/✘ |
| | Power Law | 1.49E-01/✘ | 8.41E-02/✘ | 6.03E-04/✘ | 8.07E-03/✘ | 8.31E-04/✘ |
| OPT | Exponential | 7.88E-04/✔ | 4.38E-03/✔ | 5.62E-03/✘ | 3.40E-03/✔ | 6.92E-04/✔ |
| | Power Law | 1.75E+09/✘ | 4.50E+56/✔ | 1.92E+01/✘ | 1.82E+02/✔ | 1.37E+03/✘ |
| Pythia | Exponential | 2.49E-02/✘ | 1.82E-02/✘ | 6.60E-02/✘ | 8.25E-03/✘ | 2.65E-03/✔ |
| | Power Law | 4.80E-02/✘ | 2.07E-02/✘ | 1.84E-02/✘ | 2.04E-02/✘ | 9.44E-02/✔ |

Table 15: Results from *VaCScal* corresponding to Figure 14. Numerical values, along with (✔/✘) and blue/red highlights, indicate the outcomes of Stages I, II, and III of *VaCScal*.

unusually large MoEs under both scaling models; similarly, for QAFACTEVAL, the Pythia family shows high uncertainty. We hypothesize that this arises from the uniform addition of 32 virtual tokens during Prefix-Tuning across all models within a family. This fixed augmentation likely benefits smaller models disproportionately, while offering diminishing returns for larger ones, thereby limiting their improvement in factual consistency. Despite these anomalies, exponential scaling generally results in tighter MoEs than power law scaling, indicating more stable and confident fits. The corresponding *VaCScal* results, shown in Table 15 and Table 16, reveal that although many model–dataset combinations fail feasibility in Stage II, exponential scaling consistently achieves lower held-out loss (Stage I) compared to power law scaling. Moreover, in all feasible cases, exponential scaling outperforms power law scaling in capturing the relationship between factual inconsistency and model size.

**Takeaway.** *Exponential scaling remains more stable and accurate than power law scaling in modeling factual inconsistency under Prefix-Tuning, despite some anomalies caused by uniformly applied virtual tokens—especially in smaller models like FLAN-T5 and Pythia. Even when feasibility is limited, exponential scaling consistently yields lower held-out loss and better fits in all valid cases.*

| LLM Family | Scaling | DART | E2E | ViGGO | WebNLG | WikiTableText |
|---|---|---|---|---|---|---|
| BLOOM | Exponential | 2.99E-04/✔ | 1.17E-02/✘ | 1.01E-03/✔ | 9.22E-04/✔ | 3.07E-03/✘ |
| | Power Law | 6.86E-01/✘ | 8.30E-02/✘ | 1.05E-03/✔ | 2.33E-01/✘ | 7.28E-02/✔ |
| FLAN-T5 | Exponential | 1.13E+00/✘ | 1.77E-02/✘ | 2.36E-04/✘ | 1.42E-05/✔ | 7.38E-04/✔ |
| | Power Law | 9.30E+95/✘ | 1.52E-01/✘ | 1.11E-02/✘ | 1.55E+02/✘ | 7.74E-04/✘ |
| OPT | Exponential | 7.88E-04/✔ | 4.38E-03/✔ | 5.62E-03/✘ | 3.40E-03/✔ | 6.92E-04/✔ |
| | Power Law | 1.75E+09/✘ | 4.50E+56/✔ | 1.92E+01/✘ | 1.82E+02/✔ | 1.37E+03/✘ |
| Pythia | Exponential | 1.51E-02/✘ | 2.06E-02/✘ | 9.79E-05/✘ | 2.04E-02/✘ | 2.93E-03/✘ |
| | Power Law | 1.46E+51/✘ | 3.66E-02/✘ | 4.51E+07/✘ | 2.80E-02/✘ | 1.09E+50/✘ |

Table 16: Results from *VaCScal* corresponding to Figure 15. Numerical values, along with (✔/✘) and blue/red highlights, indicate the outcomes of Stages I, II, and III of *VaCScal*.

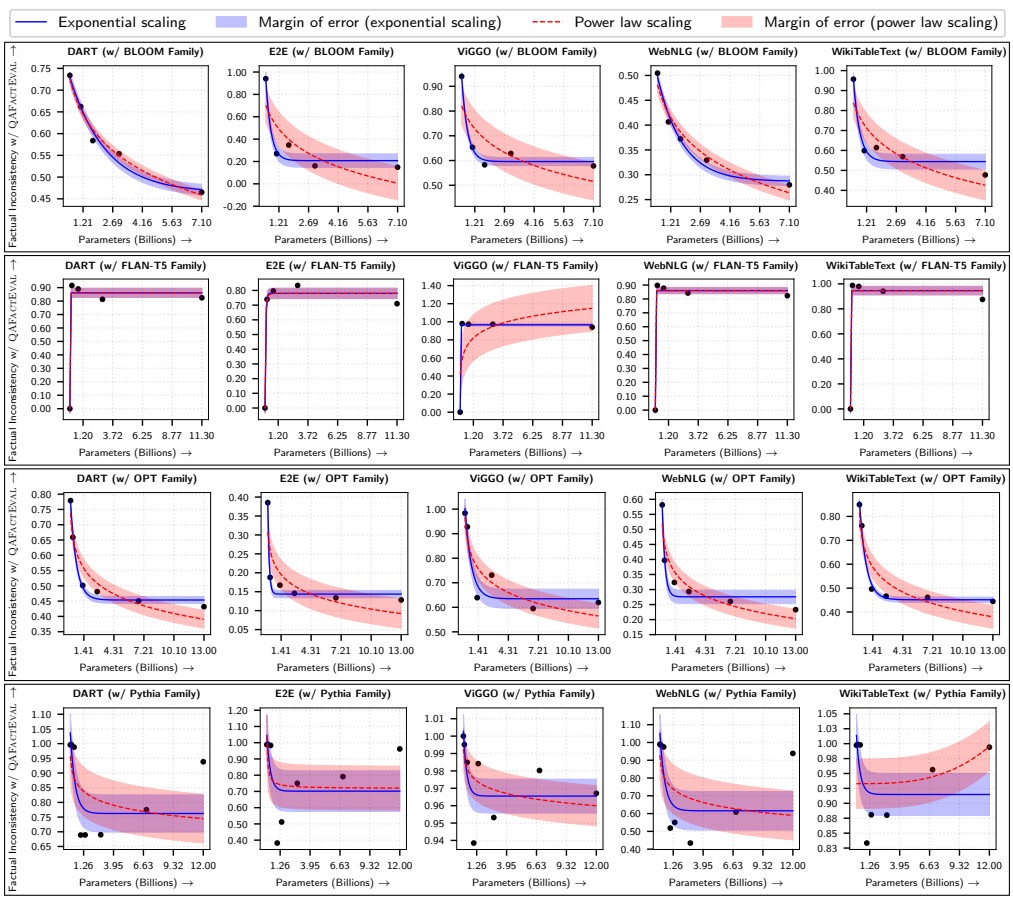

Figure 15: Fitted power law and exponential scaling models (with 95% confidence intervals) for factual inconsistency, measured by QAFACTEVAL, plotted against model size across different LLM families. Here we consider Prefix-Tuning for fine-tuning all LLM families.

# G   Result: Scaling Behavior Based on Fine-tune Data Size

Data size is a fundamental factor in understanding scaling behavior. To investigate its effect on factual inconsistency in D2T tasks with LLMs, we analyze how varying the amount of fine-tuning data influences factual inconsistency outcomes. Here, "fine-tuning data" refers to different proportions of the original training set—specifically 10%, 20%, 40%, 60%, and 100%—used during fine-tuning with the QLoRA technique. This experiment is conducted on three representative D2T datasets: DART (graph-to-text), E2E (MR-to-text), and WikiTableText (table-to-text), with factual inconsistency measured using ALIGNSCORE and QAFACTEVAL. We use the largest variants from each of the five incorporated LLM families (BLOOM, FLAN-T5, Mamba, OPT, and Pythia). Figure 16 and Figure 17 show the fitted power law and exponential scaling curves along with 95% confidence intervals (MoE), derived from residual-based estimation. Across both metrics and all model families, the E2E dataset consistently exhibits aberrant behavior—characterized by wide confidence bands and unstable scaling patterns—likely due to its inherent characteristics, which we discuss in more detail in Appendix I. DART also shows elevated MoEs, especially under ALIGNSCORE, whereas WikiTableText demonstrates relatively stable scaling behavior, with low MoEs under both models. The corresponding *VaCScal* results, presented in Table 17 and Table 18, support these observations. For the E2E dataset, neither scaling model yields feasible fits, indicating unreliable scaling behavior. However, DART (particularly under QAFACTEVAL) and WikiTableText show stronger support for exponential scaling. A few exceptions persist—for instance, the FLAN-T5 model on WikiTableText favors power law scaling, as indicated by a red-marked cell. In summary, while fine-tuning data size does not lead to universally consistent scaling across all settings, the results broadly support exponential scaling as the more robust and reliable model for capturing the relationship between factual inconsistency and data size across LLMs.

**Takeaway.** *Exponential scaling generally serves as a more robust and reliable model than power law scaling for predicting how factual inconsistency in D2T tasks decreases with increasing fine-tuning data size, despite some datasets (e.g., E2E) exhibiting aberrant or unstable behavior.*

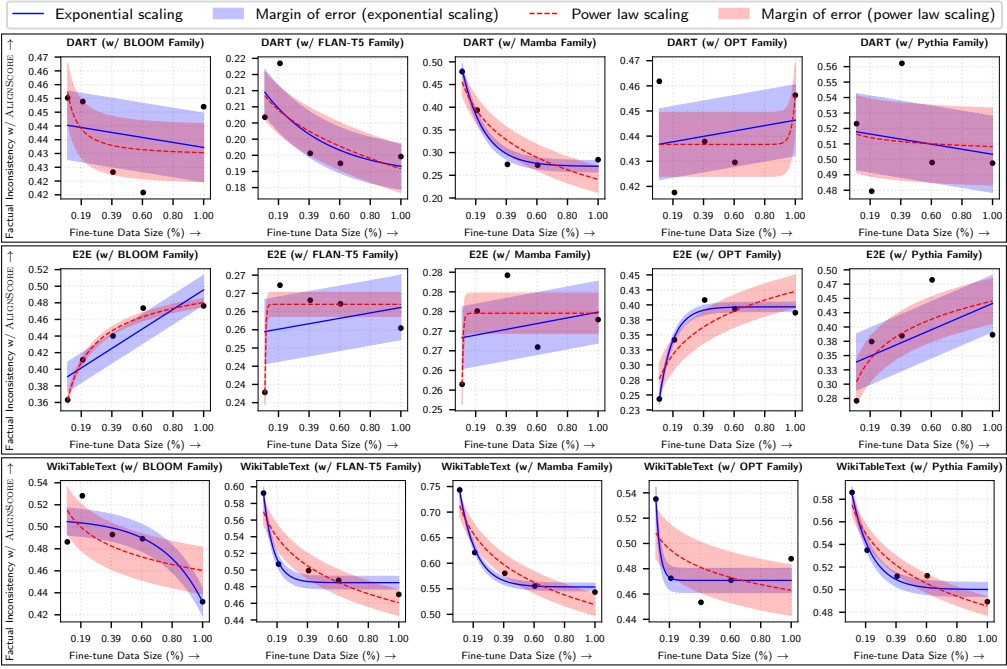

Figure 16: Fitted power law and exponential scaling models (with 95% confidence intervals) for factual inconsistency, measured by ALIGNSCORE, plotted against fine-tune data size across three datasets—DART, E2E and WikiTableText.

| Dataset | Scaling | BLOOM | FLAN-T5 | Mamba | OPT | Pythia |
|---|---|---|---|---|---|---|
| DART | Exponential | 4.17E+00/✘ | 2.00E-04/✘ | 1.20E+02/✔ | 2.95E-04/✘ | 1.39E-03/✘ |
| | Power Law | 7.72E+144/✘ | 1.24E-04/✘ | 2.90E-03/✘ | 3.16E-02/✘ | 4.60E-03/✘ |
| E2E | Exponential | 1.20E-04/✔ | 5.14E-06/✘ | 4.61E-05/✘ | 6.41E+01/✔ | 1.56E-03/✘ |
| | Power Law | 1.48E+00/✔ | 2.49E-05/✘ | 6.86E-04/✘ | 4.24E-02/✔ | 2.25E-03/✘ |
| WikiTableText | Exponential | 7.02E-04/✘ | 1.35E-03/✔ | 1.64E-03/✔ | 1.32E-04/✘ | 1.80E-04/✔ |
| | Power Law | 1.89E-01/✘ | 8.59E-05/✔ | 6.68E+00/✘ | 4.95E-03/✘ | 3.10E-02/✘ |

Table 17: Results from *VaCScal* corresponding to Figure 16. Numerical values, along with (✔/✘) and blue/red highlights, indicate the outcomes of Stages I, II, and III of *VaCScal*.

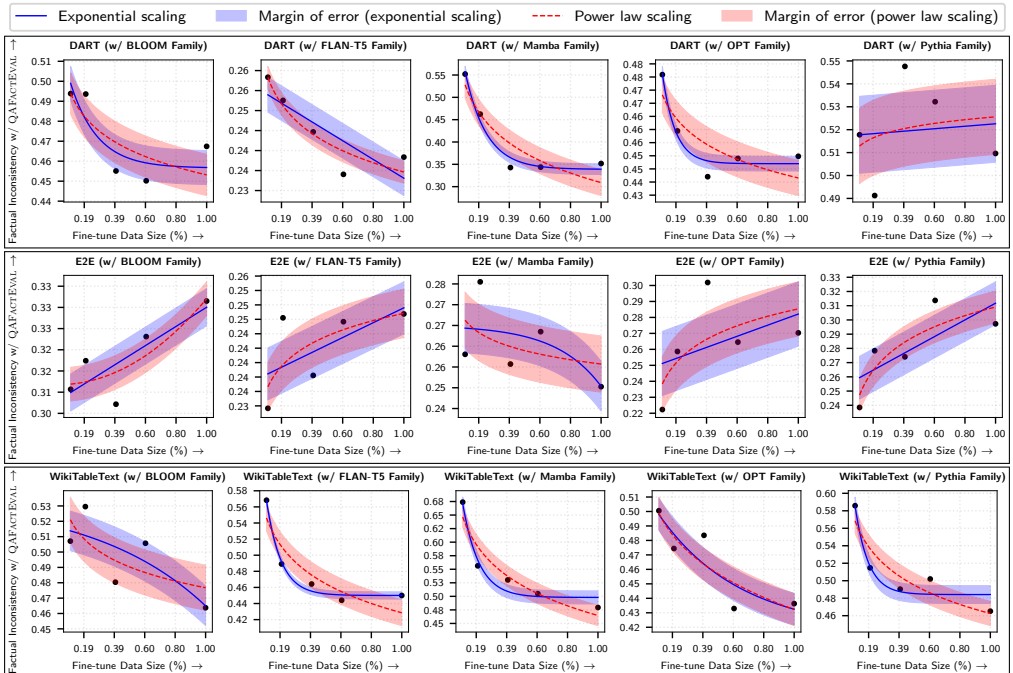

Figure 17: Fitted power law and exponential scaling models (with 95% confidence intervals) for factual inconsistency, measured by QAFACTEVAL, plotted against fine-tune data size across three datasets—DART, E2E and WikiTableText.

| Dataset | Scaling | BLOOM | FLAN-T5 | Mamba | OPT | Pythia |
|---|---|---|---|---|---|---|
| DART | Exponential | 5.83E-04/✘ | 1.53E-05/✔ | 7.41E+01/✔ | 6.25E+00/✔ | 1.52E+01/✘ |
| | Power Law | 1.74E+147/✘ | 2.87E-03/✘ | 6.66E+163/✘ | 5.70E-04/✘ | 1.23E-02/✘ |
| E2E | Exponential | 3.40E-05/✘ | 3.38E+00/✘ | 9.86E+00/✘ | 1.54E+01/✘ | 5.85E-04/✘ |
| | Power Law | 2.00E+04/✘ | 3.47E-05/✘ | 4.59E+05/✘ | 4.32E-04/✘ | 6.65E-03/✘ |
| WikiTableText | Exponential | 3.72E-04/✘ | 4.82E-04/✔ | 2.22E-03/✔ | 4.03E-04/✘ | 6.34E-04/✔ |
| | Power Law | 8.04E-04/✘ | 3.13E-04/✔ | 2.13E-04/✘ | 6.41E-02/✘ | 1.41E-02/✘ |

Table 18: Results from *VaCScal* corresponding to Figure 17. Numerical values, along with (✔/✘) and blue/red highlights, indicate the outcomes of Stages I, II, and III of *VaCScal*.

# H  Scaling Experiments with Full Fine-tuning

Unlike parameter-efficient fine-tuning methods such as QLoRA and prefix-tuning, full fine-tuning updates all model parameters, offering greater flexibility at the cost of substantially higher computational demands. Due to resource constraints, we perform full fine-tuning experiments on four LLM families—FLAN-T5, Mamba, Pythia, and Qwen2.5—across three D2T datasets: DART, E2E, and WikiTableText. Within the Qwen2.5 family, we consider only five models ranging from 0.5B to 14B parameters, as full fine-tuning of `Qwen2.5-32B` and `Qwen2.5-72B` exceeds our computational resource limits. We investigate scaling behavior with respect to both model size and computational budget (measured in FLOPs). For all outputs generated in these full fine-tuning experiments, we employ greedy decoding.

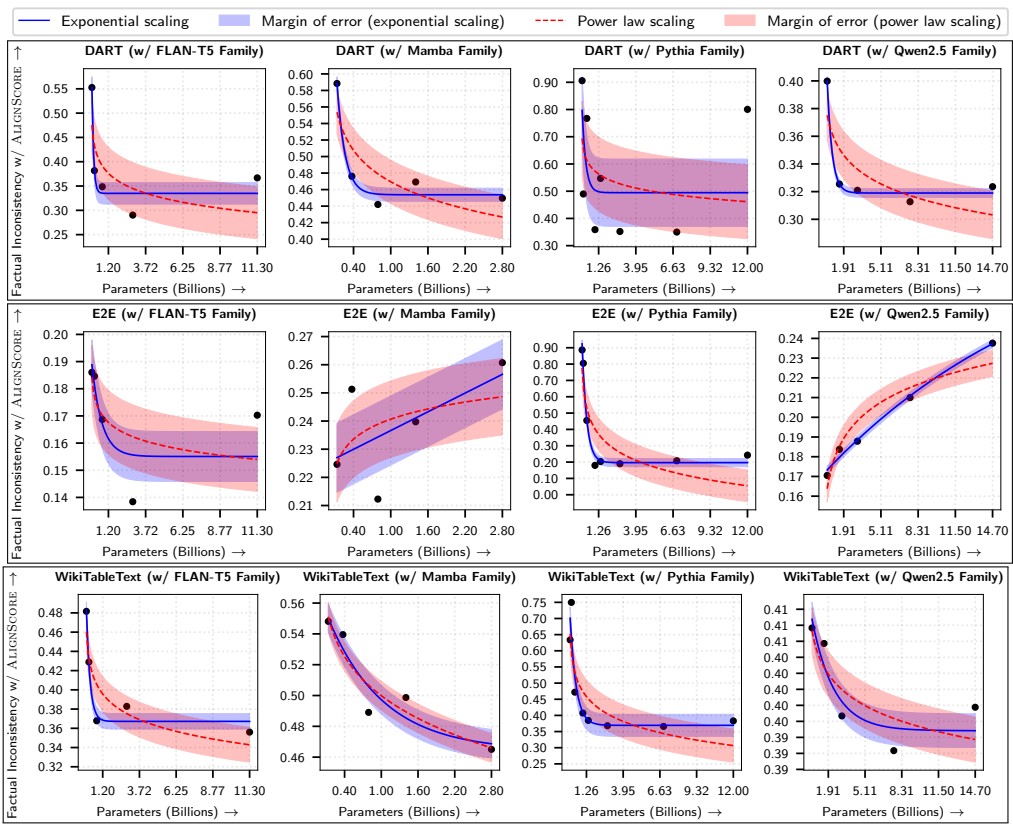

Figure 18: Fitted power law and exponential scaling models (with 95% confidence intervals) for factual inconsistency in D2T, measured by AlignScore, plotted against model size across full fine-tuned FLAN-T5, Mamba, Pythia and Qwen2.5 LLM families.

## H.1  Scaling Behavior Based on Model Size of Full Fine-Tuned LLM Families

The scaling results based on all four automatic metrics—AlignScore, QAFactEval, SummaC-conv, and UniEval-fact—across four LLM families (FLAN-T5, Mamba, Pythia, and Qwen2.5) are shown in Figure 18, Figure 19, Figure 20, and Figure 21. These results (especially in DART and WikiTableText) consistently demonstrate that exponential scaling outperforms power law scaling when model size is used as the scaling factor, as evidenced by its lower margin of error (MoE), indicating more stable and reliable fits to factual inconsistency trends in D2T tasks. This superiority is further substantiated by the outputs of the *VaCScal* framework (Table 19, Table 20, Table 21, and Table 22), which consistently favor exponential scaling over power law scaling across all metrics. Similar to our findings with QLoRA, exponential scaling under full fine-tuning also exhibits strong support in Stage III of *VaCScal* (highlighted in blue), suggesting better empirical consistency. Compared to prefix-tuning, full fine-tuning shows greater alignment with

exponential scaling. Although full fine-tuning yields fewer accepted fits in Stages II and III of *VaCScal* compared to QLoRA—exponential scaling still achieves significantly lower predictive error (Stage I) than power law scaling, reaffirming its superior predictive performance.

| Dataset | Scaling | FLAN-T5 | Mamba | Pythia | Qwen2.5 |
|---------|---------|---------|-------|--------|---------|
| DART | Exponential | 3.63E-03/✘ | 4.03E+01/✔ | 2.28E-02/✘ | 4.44E-04/✔ |
| | Power Law | 3.61E+11/✘ | 2.25E-04/✔ | 3.08E+02/✘ | 4.46E+17/✔ |
| E2E | Exponential | 6.22E-04/✘ | 2.28E+01/✘ | 6.29E-04/✔ | 1.46E-05/✔ |
| | Power Law | 1.26E-03/✘ | 2.94E-04/✘ | 3.64E+05/✘ | 1.30E-05/✔ |
| WikiTableText | Exponential | 1.22E+00/✔ | 2.61E-04/✘ | 6.12E-03/✔ | 9.80E-06/✘ |
| | Power Law | 1.21E-02/✘ | 5.18E+03/✘ | 8.45E+09/✘ | 2.43E+22/✘ |

Table 19: Results from *VaCScal* corresponding to Figure 18, along with (✔/✘) and blue/red highlights, indicate the outcomes of Stages I, II, and III of *VaCScal*.

**Takeaway.** *In the full fine-tuning setting, across all four state-of-the-art automatic metrics and LLM families (FLAN-T5, Mamba, Pythia, and Qwen2.5), exponential scaling consistently demonstrates better agreement with how factual inconsistency decreases with model size, offering a more stable and practically reliable scaling model than power law scaling.*

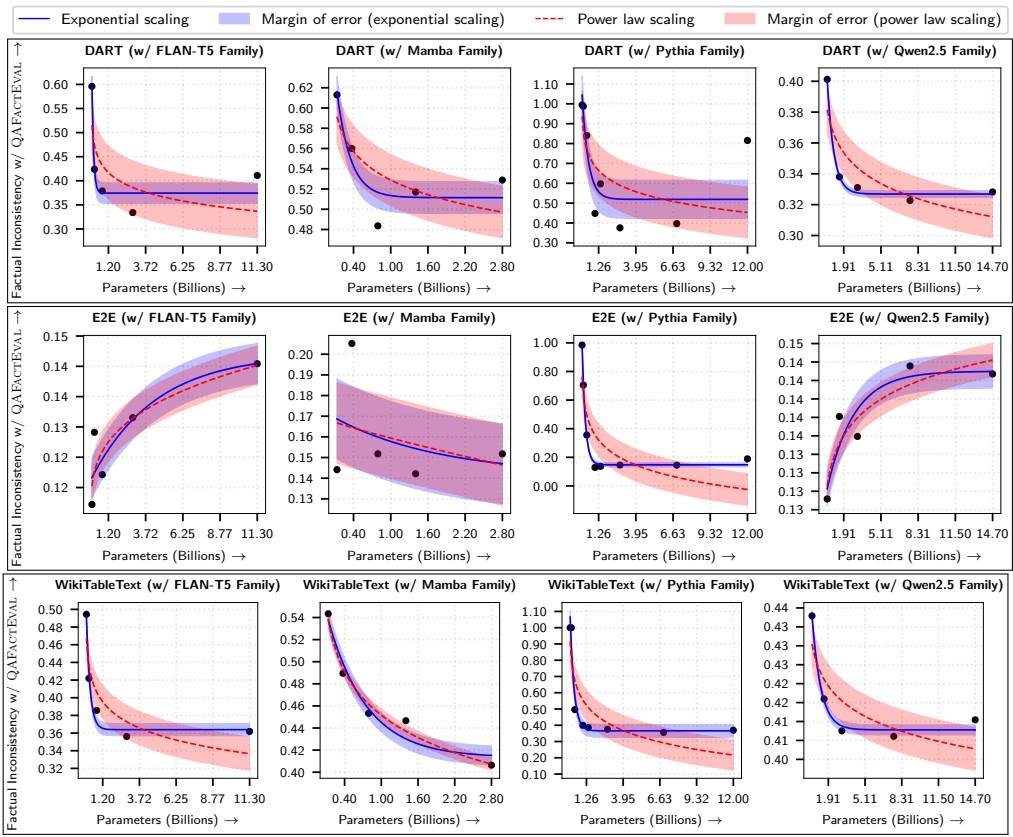

Figure 19: Fitted power law and exponential scaling models (with 95% confidence intervals) for factual inconsistency in D2T, measured by QAFACTEVAL, plotted against model size across full fine-tuned FLAN-T5, Mamba, Pythia and Qwen2.5 LLM families.

| Dataset | Scaling | FLAN-T5 | Mamba | Pythia | Qwen2.5 |
|---------|---------|---------|-------|--------|---------|
| DART | Exponential | 1.12E-03/✘ | 1.05E-03/✘ | 2.02E-02/✘ | 5.48E-04/✔ |
| | Power Law | 7.57E+19/✘ | 1.48E-03/✘ | 2.71E-02/✘ | 3.05E-04/✔ |
| E2E | Exponential | 1.60E-05/✘ | 6.69E-03/✘ | 3.63E-04/✔ | 1.38E-05/✘ |
| | Power Law | 3.76E-04/✘ | 3.23E+26/✘ | 4.55E-02/✘ | 2.49E-04/✘ |
| WikiTableText | Exponential | 9.30E-04/✔ | 3.43E-04/✔ | 2.46E-02/✔ | 1.51E+01/✔ |
| | Power Law | 1.68E-03/✘ | 9.86E-06/✘ | 5.40E+09/✘ | 3.13E-05/✔ |

Table 20: Results from *VaCScal* corresponding to Figure 19. Numerical values, along with (✔/✘) and blue/red highlights, indicate the outcomes of Stages I, II, and III of *VaCScal*.

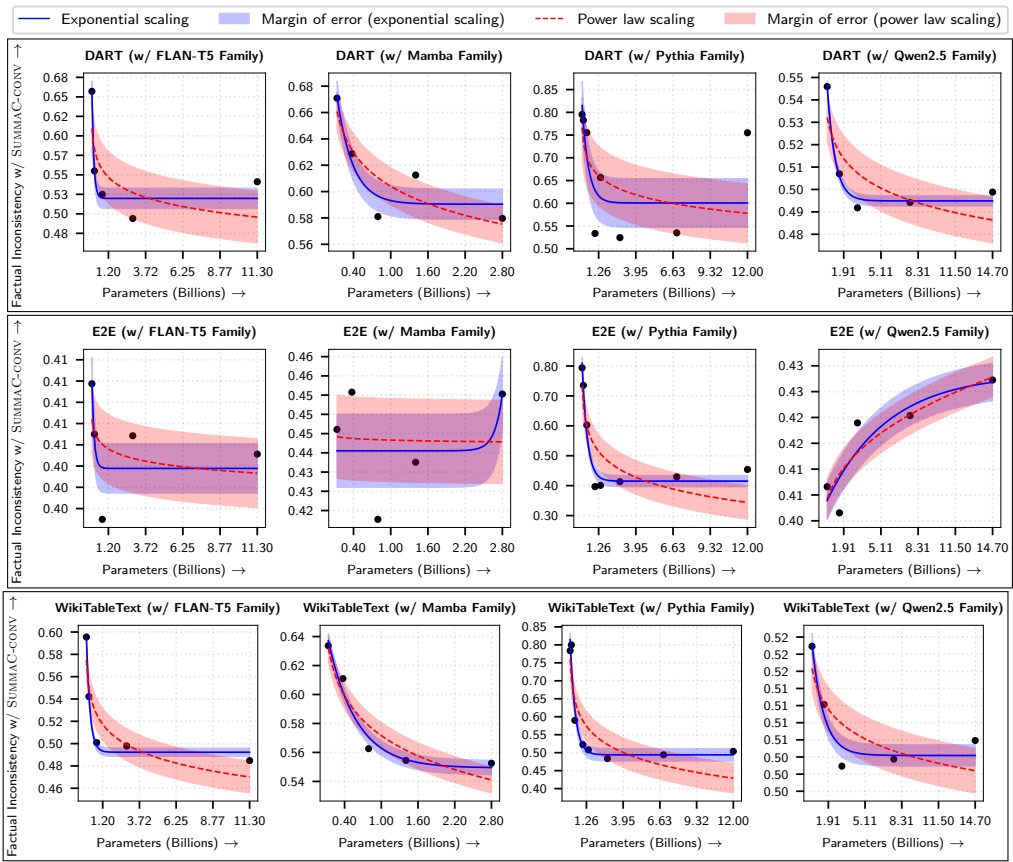

Figure 20: Fitted power law and exponential scaling models (with 95% confidence intervals) for factual inconsistency in D2T, measured by SUMMAC-CONV, plotted against model size across full fine-tuned FLAN-T5, Mamba, Pythia and Qwen2.5 LLM families.

| Dataset | Scaling | FLAN-T5 | Mamba | Pythia | Qwen2.5 |
|---------|---------|---------|-------|--------|---------|
| DART | Exponential | 1.19E-03/✔ | 6.45E-04/✘ | 5.15E-03/✘ | 1.18E+02/✔ |
| | Power Law | 1.23E-03/✘ | 4.38E-04/✘ | 1.38E+00/✘ | 6.58E-05/✔ |
| E2E | Exponential | 2.03E-05/✘ | 3.17E+01/✘ | 1.08E-03/✔ | 2.07E-04/✘ |
| | Power Law | 2.76E-06/✘ | 5.18E-04/✘ | 9.26E+04/✘ | 8.35E-03/✘ |
| WikiTableText | Exponential | 3.79E-05/✔ | 1.34E-03/✔ | 3.26E-03/✔ | 3.17E+01/✔ |
| | Power Law | 1.26E-02/✘ | 8.00E+14/✘ | 6.00E+06/✘ | 1.03E-04/✘ |

Table 21: Results from *VaCScal* corresponding to Figure 20. Numerical values, along with (✔/✘) and blue/red highlights, indicate the outcomes of Stages I, II, and III of *VaCScal*.

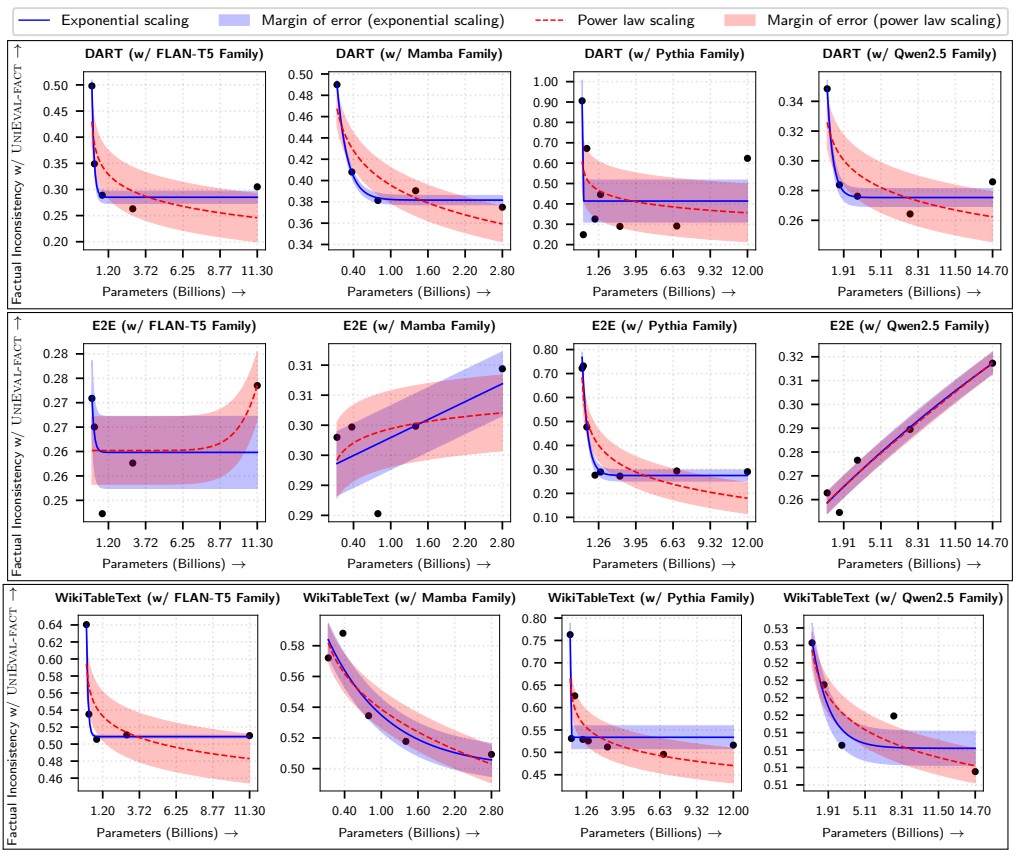

Figure 21: Fitted power law and exponential scaling models (with 95% confidence intervals) for factual inconsistency in D2T, measured by UNiEVAL-FACT, plotted against model size across full fine-tuned FLAN-T5, Mamba, Pythia and Qwen2.5 LLM families.

| Dataset | Scaling | FLAN-T5 | Mamba | Pythia | Qwen2.5 |
|---------|---------|---------|-------|--------|---------|
| DART | Exponential | 1.17E-03/✔ | 2.49E+01/✔ | 6.10E-02/✘ | 2.45E-04/✔ |
| | Power Law | 2.40E+19/✘ | 3.64E-03/✔ | 4.99E+89/✘ | 2.03E-04/✔ |
| E2E | Exponential | 1.43E-04/✘ | 9.19E-05/✘ | 1.79E-03/✔ | 1.06E-05/✔ |
| | Power Law | 1.02E-03/✘ | 1.36E-03/✘ | 5.24E+136/✘ | 3.18E-05/✔ |
| WikiTableText | Exponential | 1.28E-04/✔ | 9.08E-04/✘ | 3.63E-04/✔ | 1.02E+01/✘ |
| | Power Law | 1.32E-04/✔ | 1.13E-03/✘ | 1.38E-03/✔ | 3.04E+134/✘ |

Table 22: Results from *VaCScal* corresponding to Figure 21. Numerical values, along with (✔/✘) and blue/red highlights, indicate the outcomes of Stages I, II, and III of *VaCScal*.

## H.2 Scaling Behavior Based on FLOPs of Full Fine-Tuned LLM Families

We present the scaling behavior of four LLM families—FLAN-T5, Mamba, Pythia, and Qwen2.5—on three D2T datasets (DART, E2E, and WikiTableText) based on model size, using two automatic factual consistency metrics: AlignScore and QAFactEval. Our analysis reveals that the scaling behavior with respect to FLOPs under full fine-tuning closely mirrors the trends observed with model size. As shown in Figure 22 and Figure 23, exponential scaling consistently outperforms power law scaling across all four LLM families in terms of lower margin of error (MoE), indicating more stable and reliable fits. The outcomes from the *VaCScal* framework (Table 23 and Table 24) further reinforce the superiority of exponential scaling, offering strong empirical support when FLOPs are considered as the scaling factor.

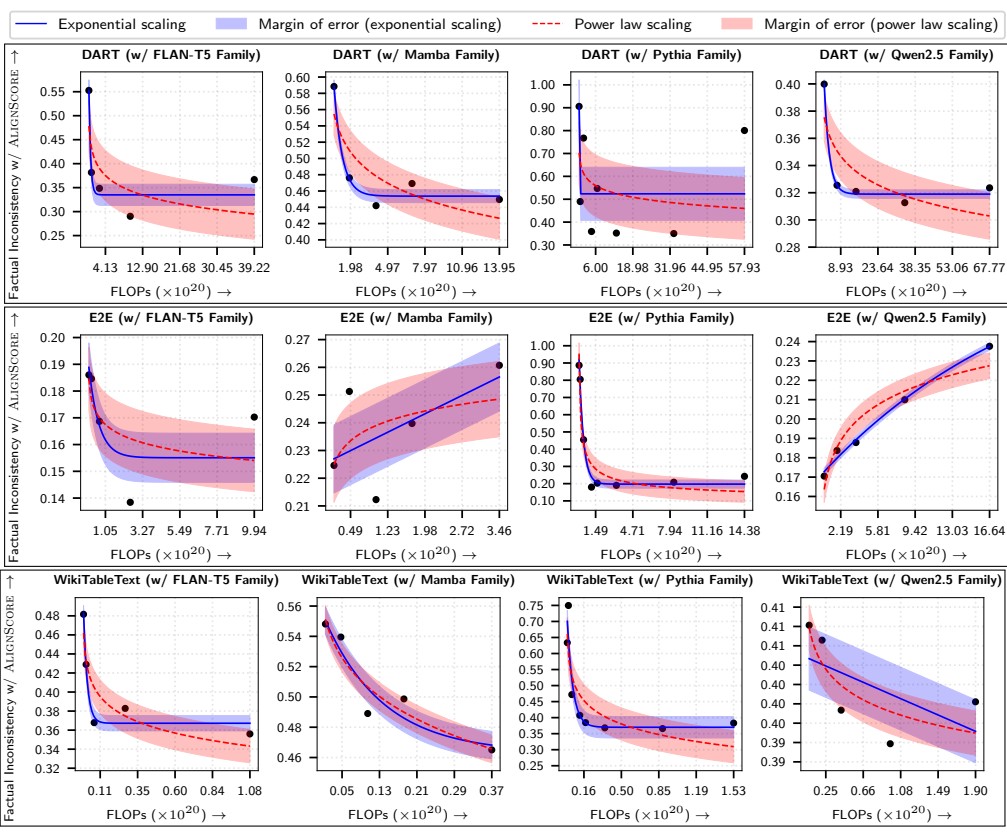

Figure 22: Fitted power law and exponential scaling models (with 95% confidence intervals) for factual inconsistency in D2T, measured by AlignScore, plotted against FLOPs across full fine-tuned FLAN-T5, Mamba, Pythia and Qwen2.5 LLM families.

| Dataset | Scaling | FLAN-T5 | Mamba | Pythia | Qwen2.5 |
|---------|---------|---------|-------|--------|---------|
| DART | Exponential | 3.36E-03/✘ | 1.96E-04/✔ | 8.98E+01/✘ | 4.58E-05/✔ |
| | Power Law | 3.30E+11/✘ | 8.52E-04/✔ | 4.16E-01/✘ | 4.65E-05/✔ |
| E2E | Exponential | 2.42E-05/✘ | 1.15E+01/✘ | 2.87E-03/✔ | 1.19E-05/✔ |
| | Power Law | 2.27E-04/✘ | 5.22E-04/✘ | 1.71E+137/✘ | 1.13E-05/✔ |
| WikiTableText | Exponential | 3.39E+00/✔ | 3.06E-04/✘ | 6.08E-03/✔ | 8.81E-05/✘ |
| | Power Law | 1.23E-02/✘ | 6.19E+03/✘ | 4.28E+07/✘ | 7.24E+22/✘ |

Table 23: Results from *VaCScal* corresponding to Figure 22. Numerical values, along with (✔/✘) and blue/red highlights, indicate the outcomes of Stages I, II, and III of *VaCScal*.

**Takeaway.** *In the full fine-tuning setting, across all four state-of-the-art automatic metrics and LLM families (FLAN-T5, Mamba, Pythia, and Qwen2.5), exponential scaling shows stronger acceptance in modeling*

*the decline of factual inconsistency in D2T—with respect to FLOPs—based on ALIGNSCORE and QAFACTE-VAL, offering a more stable and practically reliable scaling model compared to power law scaling.*

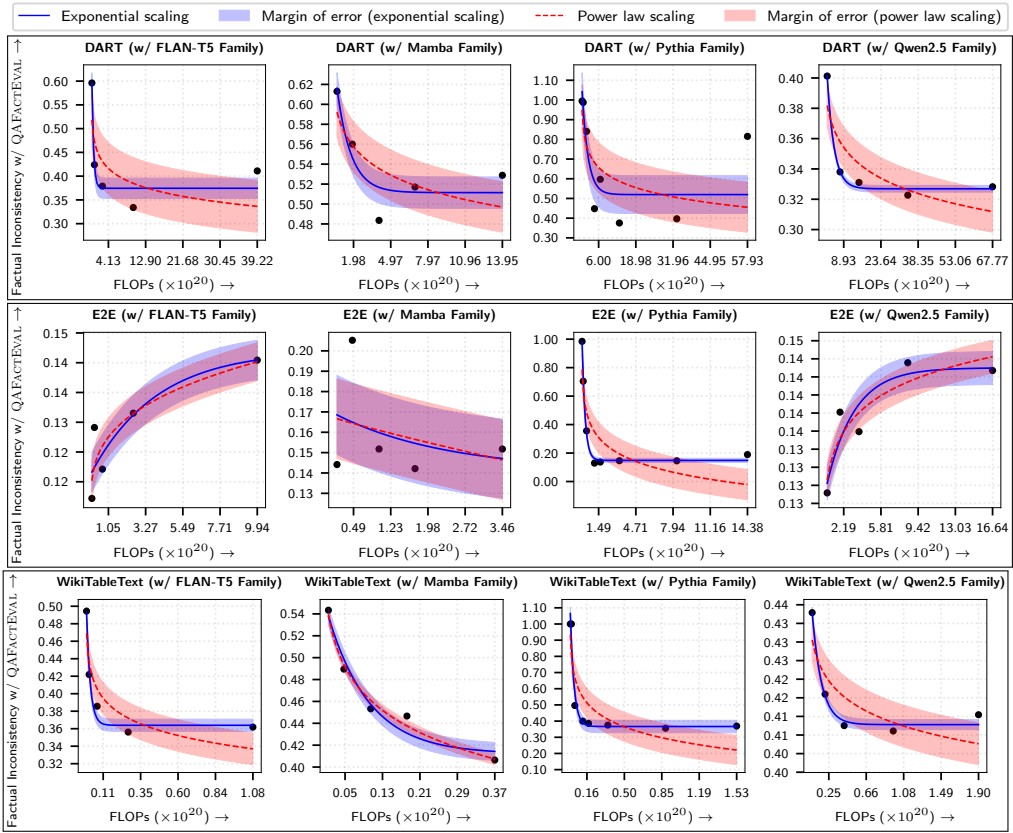

Figure 23: Fitted power law and exponential scaling models (with 95% confidence intervals) for factual inconsistency in D2T, measured by QAFACTEVAL, plotted against FLOPs across full fine-tuned FLAN-T5, Mamba, Pythia and Qwen2.5 LLM families.

| Dataset | Scaling | FLAN-T5 | Mamba | Pythia | Qwen2.5 |
|---------|---------|---------|-------|--------|---------|
| DART | Exponential | 3.63E-03/✖ | 4.82E-04/✖ | 1.86E-02/✖ | 3.23E-05/✔ |
| | Power Law | 1.47E-03/✖ | 5.92E-04/✖ | 2.84E-02/✖ | 5.59E-05/✔ |
| E2E | Exponential | 2.24E-05/✖ | 1.54E-04/✖ | 1.59E+02/✔ | 1.03E-05/✖ |
| | Power Law | 3.56E-04/✖ | 1.22E+27/✖ | 3.53E+133/✖ | 7.00E-06/✖ |
| WikiTableText | Exponential | 8.80E-05/✔ | 3.81E-04/✔ | 2.52E-02/✔ | 2.63E-04/✔ |
| | Power Law | 6.07E-03/✖ | 1.51E-01/✖ | 1.17E+09/✔ | 1.08E-05/✔ |

Table 24: Results from *VaCScal* corresponding to Figure 23. Numerical values, along with (✔/✖) and blue/red highlights, indicate the outcomes of Stages I, II, and III of *VaCScal*.

# I  On the Aberrant Scaling Patterns of E2E and ViGGO

Throughout our experiments, we consistently observed aberrant behavior from the E2E dataset, and to a lesser extent, the ViGGO dataset. Specifically, factual inconsistency in these datasets tends to increase with scaling factors such as model size, FLOPs, and fine-tuning data size—contrary to expected trends. Although we could not definitively identify the root cause of this behavior, we hypothesize that the low lexical diversity and closed-domain nature of these datasets may be contributing factors. Limited diversity can lead to overfitting during fine-tuning, resulting in degraded output quality and increased factual inconsistency. Additionally, the closed-domain nature of the E2E and ViGGO datasets (see Table 6) may hinder LLMs from effectively leveraging their pre-trained knowledge. To support this hypothesis, we present the type-token ratio (TTR), computed using the following formula, for all five datasets used in our study:

$$\text{TTR} = \frac{\text{number of unique words or tokens}}{\text{total number of words and tokens}}$$

A lower TTR indicates reduced lexical diversity. As shown in Table 25, the E2E and ViGGO datasets exhibit the lowest lexical richness—both in the input source data and the corresponding references—compared to the other three datasets.

|  | DART | E2E | ViGGO | WebNLG | WikiTableText |
|---|---|---|---|---|---|
| TTR (of source-input) | 0.0178 | 0.0001 | 0.0029 | 0.0057 | 0.0621 |
| TTR (of reference text) | 0.03 | 0.0058 | 0.0297 | 0.0209 | 0.1297 |

Table 25: Type-Token Ratio (TTR) for all five D2T datasets, calculated separately for input source data and corresponding reference texts.

## J  Mathematical Interpretation of Exponential Scaling in Factual Inconsistency for D2T with LLM Size

To develop an intuitive understanding of why factual inconsistency in D2T generation often exhibits exponential scaling with LLM size, we draw on a few mathematical insights. We hypothesize that one of the primary causes of factual inconsistency in D2T tasks is *source-reference divergence*—a discrepancy between the input source data and the reference outputs in the training (or fine-tuning) data. Such divergence introduces systematic biases during learning, which manifest as factual errors at inference time. Consequently, standard perplexity—computed against the training reference—fails to serve as a reliable indicator of factual inconsistency, since the model is trained on references that may themselves contain factual deviations. To address this limitation, we propose a relative perplexity measure, which more closely (though NOT exactly) reflects the degree of factual inconsistency. Using relative perplexity instead of ordinary perplexity is motivated by the need to compare how the model scores a factual reference against its own most likely (and potentially inconsistent) output. While ordinary perplexity captures the overall fluency or likelihood of a sequence, it doesn't reveal whether the model prefers a hallucinated output over the truth. Formally, we define the relative perplexity $\mathcal{R}$ between the input $x$ and the true factual reference $y^{\mathrm{ref}} = y_1^{\mathrm{ref}} y_2^{\mathrm{ref}} \ldots y_L^{\mathrm{ref}}$ as the ratio between the perplexity of the true reference and that of the model's most likely output at inference, $y^{\mathrm{max}} = y_1^{\mathrm{max}} y_2^{\mathrm{max}} \ldots y_L^{\mathrm{max}}$, where $L$ denotes the sequence length of both texts (Equation 6). Note that the "true" reference here may differ from the dataset-provided reference; it may also incorporate information directly grounded in the input $x$.

$$\mathcal{R}(x, y^{\mathrm{ref}}) = \left. \frac{\mathrm{perplexity}(y^{\mathrm{ref}})}{\mathrm{perplexity}(y^{\mathrm{max}})} \right|_{\mathrm{input}=x} = \left. \left( \frac{\prod_{i=1}^{L} \mathrm{Pr}(y_i^{\mathrm{ref}})}{\prod_{i=1}^{L} \mathrm{Pr}(y_i^{\mathrm{max}})} \right)^{-\frac{1}{L}} \right|_{\mathrm{input}=x} \tag{6}$$

The relative perplexity $\mathcal{R}$ takes values in the range $[1, \infty)$, where a value of 1 indicates minimal factual inconsistency—i.e., the LLM assigns maximum likelihood to the true factual reference. As $\mathcal{R}$ increases (i.e., $\mathcal{R} \to \infty$), it reflects growing factual inconsistency between the model's output and the true reference. In practice, this range may be bounded through proper calibration of the LLM.

Since most LLMs compute token probabilities via a softmax function applied over logits at inference time, we now turn our attention to the logits underlying the tokens in Equation 6. Let $z^{\mathrm{ref}}$ and $z^{\mathrm{max}}$ denote the logits associated with $y^{\mathrm{ref}}$ and $y^{\mathrm{max}}$, respectively.

$$\mathcal{R}(x, y^{\mathrm{ref}}) = \left. \left( \frac{\prod_{i=1}^{L} \frac{e^{z_i^{\mathrm{ref}}}}{\sum_j e^{z_{i,j}^{\mathrm{ref}}}}}{\prod_{i=1}^{L} \frac{e^{z_i^{\mathrm{max}}}}{\sum_j e^{z_{i,j}^{\mathrm{max}}}}} \right)^{-\frac{1}{L}} \right|_{\mathrm{input}=x} \overset{=}{_{(1)}} \left. \left( \frac{\prod_{i=1}^{L} e^{z_i^{\mathrm{ref}}}}{\prod_{i=1}^{L} e^{z_i^{\mathrm{max}}}} \right)^{-\frac{1}{L}} \right|_{\mathrm{input}=x} \overset{=}{_{(2)}} \left. \left( \prod_{i=1}^{L} e^{z_i^{\mathrm{ref}} - z_i^{\mathrm{max}}} \right)^{-\frac{1}{L}} \right|_{\mathrm{input}=x} \tag{7}$$

In the first (1) case of Equation 7, the simplification follows directly from the observation that the softmax denominators in both the numerator and the denominator are identical and hence cancel out. It is crucial to emphasize that the terms $z_i^{\mathrm{ref}}$ and $z_i^{\mathrm{max}}$ do not denote the $i$-th index in the vocabulary $V$. Instead, they refer to the logits assigned to the $i$-th token in the respective sequences $y^{\mathrm{ref}}$ and $y^{\mathrm{max}}$. Interpreting Equation 7, we see that the relative perplexity $\mathcal{R}$—which we propose as a proxy for factual inconsistency—captures the aggregate deflection of the logits assigned to the factual reference tokens from those assigned to the model's most likely output. Given that D2T datasets often exhibit source-reference divergence, it is expected that such a mismatch between $z_i^{\mathrm{ref}}$ and $z_i^{\mathrm{max}}$ will frequently occur. In the following part, we mathematically analyze the nature and implications of this mismatch in greater detail.

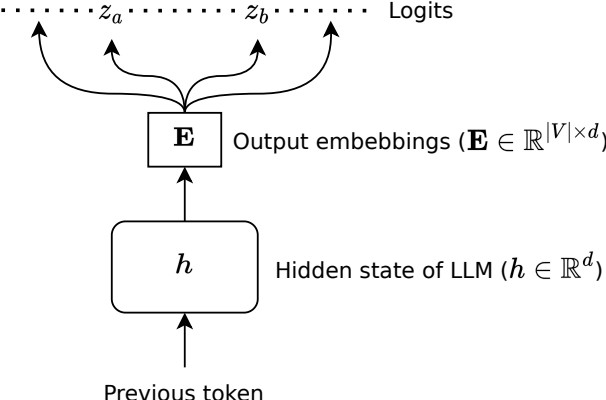

Figure 24: A simplified systematic view of the logits output in a LLM.

For notational simplicity, let us denote $z_a = z_i^{\text{ref}}$ and $z_b = z_i^{\text{max}}$, where $a$ and $b$ are the indices of the corresponding tokens in the vocabulary $V$. Before proceeding further, we make the following assumptions about the generation of logits. At a time step, the logit vector is computed as $z = \mathbf{E}h$, where $\mathbf{E} \in \mathbb{R}^{|V| \times d}$ is the output embedding matrix (mapping hidden representations to vocabulary logits), and $h \in \mathbb{R}^d$ is the hidden state of the LLM at time step $t$. Assume the model dimension is $d$, i.e., $\dim(h) = d$. Following training and under a normalized regime, we assume that the elements of $\mathbf{E}$ and $h$ are independent and identically distributed (IID) with zero mean: $\mathbf{E}_{i,j} \sim \mathcal{N}(0, \sigma_E^2)$ and $h_j \sim \mathcal{N}(0, \sigma_h^2)$ for all valid indices $i, j$, and all random variables are mutually independent. Under these assumptions, the logit value for the $k$-th token in the vocabulary is given by following Equation 8.

$$z_k = \sum_{j=1}^{d} \mathbf{E}_{k,j} h_j \tag{8}$$

The variance of any logit can be computed using the expression for $z_k$ from Equation 8, as follows:

$$\text{var}[z_k] = d \cdot \sigma_h^2 \cdot \sigma_E^2 \tag{9}$$
$$= d \cdot \sigma^2 \qquad \text{Assuming } \sigma^2 = \sigma_h^2 \cdot \sigma_E^2 \tag{10}$$

It is well established in prior work on scaling laws (Sharma & Kaplan, 2022; Bahri et al., 2024) that the model dimension $d$ scales with the model size $N$ according to a function $\phi$, i.e., $d \propto \phi(N)$, where $\phi$ is typically linear or sub-linear. Given this relationship, we now aim to estimate the expected mismatch between the logits $z_a$ and $z_b$. To that end, we invoke Chebyshev's inequality to bound the probability of large deviations between these two logits. This provides a principled estimate of the likelihood of factual inconsistency as captured by the logit-level difference $|z_a - z_b|$.

$$\Pr\left(|z_a - z_b| \geq \xi(N)\right) \leq \frac{\text{Var}(z_a - z_b)}{\xi(N)^2} \tag{11}$$

In Equation 11, we define $\xi(N)$ to be a linear or sub-linear function of the model size $N$. Now, let us recall the basic rule for the variance of the difference between two random variables. Given two independent random variables, the variance of their difference is given by:

$$\text{Var}(z_a - z_b) = \text{Var}(z_a) + \text{Var}(z_b) - 2\,\text{Cov}(z_a, z_b) \tag{12}$$

We now observe that $z_a = \mathbf{E}_{a,:}^\top \mathbf{h}$ and $z_b = \mathbf{E}_{b,:}^\top \mathbf{h}$, where $\mathbf{E}_{a,:}$ and $\mathbf{E}_{b,:}$ denote the $a$-th and $b$-th rows of the output embedding matrix $\mathbf{E}$, respectively. Under this formulation, the covariance between $z_a$ and $z_b$ can be expressed as:

$$\text{Cov}(z_a, z_b) = \mathbf{E}_{a,:}^\top \text{Cov}(\mathbf{h})\mathbf{E}_{b,:} \tag{13}$$
$$= \sigma_h^2 \cdot \mathbf{E}_{a,:}^\top \mathbf{E}_{b,:} \tag{14}$$

An important observation here is that the term $\mathbf{E}_{a,:}^\top \mathbf{E}_{b,:}$ corresponds to the inner product (or similarity) between the embeddings of the $a$-th and $b$-th vocabulary tokens. When the embeddings are normalized, this similarity lies in the range $[-1, 1]$. According to Equation 14, the variance of the logit difference can then be expressed as $\text{Var}(z_a - z_b) = \mathcal{O}(d \cdot \sigma^2)$, where $\sigma^2$ denotes the product of variances $\sigma_E^2 \sigma_h^2$, and $d \propto \phi(N)$ for some linear or sub-linear function $\phi$ of the model size $N$. This variance becomes particularly large when $\mathbf{E}_{a,:}$ and $\mathbf{E}_{b,:}$ are nearly orthogonal (i.e., dissimilar), which is frequently the case in D2T tasks involving divergent source-reference training pairs—where the model must learn to associate inputs and references that contain distinct or conflicting facts. Now, applying Chebyshev's inequality from Equation 11, we obtain the upper bound $\frac{\text{Var}(z_a - z_b)}{\xi(N)^2}$, where $\xi(N)$ is assumed to be a linear or sub-linear function of $N$. Since $\text{Var}(z_a - z_b) = \mathcal{O}(d \cdot \sigma^2)$, the right-hand side can still be a relatively large quantity for realistic values of $N$ and $\xi(N)$. This implies that the event $|z_a - z_b| \geq \xi(N)$ is not unlikely—in fact, it may occur with high probability under typical D2T conditions. Consequently, the occurrence of significant logit deviations between the factual reference and the maximum-likelihood output is a plausible and theoretically grounded phenomenon.

Recall that $z_a = z_i^{\text{ref}}$ and $z_b = z_i^{\text{max}}$. Therefore, the probability of observing a deviation of the form $z_i^{\text{ref}} - z_i^{\text{max}} \geq \xi(N)$ is also high. Substituting $\xi(N)$ in place of $z_i^{\text{ref}} - z_i^{\text{max}}$ into the expression in Equation 7, we obtain:

$$\mathcal{R}(x, y^{\text{ref}}) = \left( \prod_{i=1}^{L} e^{\xi(N)} \right)^{-\frac{1}{L}} \Bigg|_{\text{input}=x} = e^{-\xi(N)} \Big|_{\text{input}=x} \tag{15}$$

Hence, based on the above reasoning and Equation 15, we can conclude that the relative perplexity $\mathcal{R}$—which serves as a proxy (NOT an exact measure) for factual inconsistency—tends to scale exponentially with the size of the LLM, denoted by $N$. Thus, generalizing Equation 15, we obtain:

$$\text{Factual Inconsistency} \propto e^{-\xi(N)} \Big|_{\text{input}=x} \tag{16}$$

where $\xi(N)$ is a linear or sub-linear function of the model size $N$.

This conclusion follows from Equation 16, which implies that as the LLM size $N$ increases, the logit deviations between $z_i^{\text{ref}}$ and $z_i^{\text{max}}$ also grow. When aggregated within the relative perplexity $\mathcal{R}$—used here as a proxy for factual inconsistency—these deviations exhibit exponential dependence on $N$. Therefore, under the presence of source-reference divergence in D2T datasets, factual inconsistency is expected to follow an exponential scaling pattern with respect to model size.

