# OpenReview forum: "Exponential Scaling of Factual Inconsistency in Data-to-Text Generation with Fine-Tuned LLMs"
_TMLR — Accepted by TMLR_

### Review · Reviewer_YPVG · 2025-06-29

**Summary Of Contributions:**

his paper investigates the scaling laws in the field of data-to-text generation (D2T) by examining the relationship between factual inconsistency and the size of large language models (LLMs).

The authors conduct an extensive empirical analysis to explore how factual inconsistency in D2T tasks scales with the size of LLMs. They compare two scaling laws—power law and exponential scaling—across three major LLM families (Pythia, OPT, and BLOOM) and five well-established D2T datasets (E2E, ViGGO, WebNLG, DART, and WikiTableText).

The authors introduce a structured three-stage statistical validation framework to rigorously evaluate the scaling laws:

1. Predictive Performance Estimation: Assessing the predictive ability of the scaling laws on unseen data using Huber loss.
2. Goodness-of-Fit Assessment: Evaluating how well the scaling laws fit the data using an F-test.
3. Comparative Analysis: Comparing the power law and exponential scaling models using Vuong’s likelihood-ratio test to determine which model better explains the data.

The study reveals that factual inconsistency in D2T follows an exponential scaling with LLM size, rather than the commonly assumed power law scaling. This finding is validated through extensive empirical results and the rigorous statistical framework. The exponential scaling indicates a rapid initial decline in factual inconsistency up to approximately 3–4 billion parameters, after which it stabilizes.

**Audience:**

Yes

**Broader Impact Concerns:**

No need

**Claims And Evidence:**

Yes

**Requested Changes:**

1. Conduct Experiments with Full Parameter Fine-Tuning

2. Utilize Stronger Base Models like Llama or Qwen

3. Perform More Extensive Experimental Analysis

**Strengths And Weaknesses:**

Strengths:

1. The study provides a thorough empirical investigation of the relationship between factual inconsistency and LLM size across multiple datasets and models. This comprehensive approach offers valuable insights into the behavior of LLMs in D2T tasks.

2. The authors employ a structured three-stage statistical validation framework, which includes predictive performance estimation, goodness-of-fit assessment, and comparative analysis.

3. The research addresses a critical issue in D2T—factual inconsistency—which is highly relevant for ensuring trustworthiness in applications such as automated journalism and conversational systems. The findings that factual inconsistency scales exponentially with LLM size are novel and provide new directions for future research.

Weaknesses:

1. The use of QLoRA for fine-tuning LLMs may invalidate the meaning of the scaling factor  x in the scaling law. In theory, full parameter fine-tuning would better observe model performance.

2. Some newer popular LLMs, such as Llama and Qwen, should be included in the experiments. These models often have stronger comprehension abilities. It remains to be seen whether stronger base models still follow exponential scaling.

3. The authors primarily use automatic metrics to observe performance, which can introduce more uncertainty. Although they also include human evaluation, the scale is relatively small.

4. After the model size exceeds 2 billion parameters, factual inconsistency decreases sharply, but the authors do not analyze this phenomenon.

---

> ### Author Response · Authors · 2025-08-06
>
> We sincerely thank you for your thorough review, insightful feedback, and constructive suggestions.
>
>
> ### Response to "Weaknesses"
>
> > The use of QLoRA for fine-tuning LLMs may invalidate the meaning of…
>
> Thank you for this important suggestion. In the revised version, we have included experiments using the full fine-tuning approach, which are presented in Appendix H. In full fine-tuning, as with QLoRA, we also observe a clear dominance of exponential scaling over power law scaling in modeling factual inconsistency.
>
> > Some newer popular LLMs, such as Llama and Qwen, should be included…
>
> Thank you for your suggestion. We are pleased to inform you that the Qwen2.5 family has been incorporated into the revised version of our paper. In the Qwen2.5 family, as with most of the other incorporated LLM families, we also observe a clear dominance of exponential scaling over power law scaling in modeling factual inconsistency.
>
> > ...Although they also include human evaluation, the scale is relatively small.
>
> In addition to our earlier human evaluation on OPT, we have now extended the evaluation to include FLAN-T5, Mamba, and Qwen2.5, as presented in Appendix B.
>
> > After the model size exceeds 2 billion parameters, factual inconsistency decreases sharply...
>
> Thank you for raising this point. However, our observations indicate that factual inconsistency in D2T does not consistently decrease sharply beyond the 2B-parameter mark across all LLM families. For instance, some models—such as FLAN-T5—show notable improvements in factual consistency at much smaller scales, whereas others like BLOOM require significantly larger sizes. This suggests that the exponential decline in inconsistency is highly model-dependent. In the revised version, Section 8 (with details in Appendix J) provides a mathematical perspective on this phenomenon by interpreting factual inconsistency as a function of relative perplexity. There, we specifically argue that the presence of source-reference divergence in training data may contribute to the observed prominence of exponential scaling in D2T tasks. We hope this mathematical insight will be helpful in addressing this matter.
>
> ### Response to "Requested Changes"
>
> We have already addressed this point in our response to the “Weaknesses” section.
>
> Thank you once again for your insightful feedback. Please let us know if you have any further questions or require additional clarification; we would be glad to address them.

---

### Review · Reviewer_jHSa · 2025-07-02

**Summary Of Contributions:**

This work examines scaling laws for different kinds of datasets and model families in the area of data-to-text generation in the context of factual inconsistency.
In the same vein, it proposes an extended methodology to determine and validate scaling laws through a three-step procedure called VaCScal, which makes use tof the F-test to assess goodness-of-fit and Vuong's likelihood ratio test to decide whether to prefer a exponential or power law scaling model.
They find that exponential scaling better explains factual inconsistency as a function of model size, FLOPs and finetuning dataset set size across families and factuality metrics.

**Audience:**

Yes

**Broader Impact Concerns:**

I foresee no broader impact concerns with this paper.

**Claims And Evidence:**

Yes

**Requested Changes:**

I have only very few requested changes with the structure of the paper, as I think the content itself is very good.

* The scaling law figures (figure 3-6) could be a bit scaled up for readability. I acknowledge that this was probably done in order to preserve space. Therefore, I also wonder whether authors could show only some of the mentioned figures in the main paper and move figures with similar conclusions / trends to the appendix, or only show the scaling laws for some datasets in the main text. This would also decrease the amount of information a reader has to parse at once.
* It is also apparent to me that due to giving the presentation and discussion of experimental results a lot of space, many other relevant sections had to be moved to the appendix. Sections I would personally consider relevant to the main text are for instance section A.1 that explains the kinds of dataset used, appendix F, and at least a short summary of the discussion in appendix G regarding the counter-intuitive "anti-scaling" results for some datasets.
* The same holds for the derivation in Appendix H - the result could be presented in abbreviated / simplified form, with the detailed derivation in the appendix.

Other minor points:
* Missing citations for RoBERTa and Reddit / Pile datasets at the top of page 17
* I believe the comma after "two scaling models" in Stage III on page 4 is redundant.

**Strengths And Weaknesses:**

Strengths
-----------
* The paper is very polished and clearly written.
* Results are presented clearly and comprehensively.
* The extension of scaling laws with VaCScal is (to the best of my knowledge) novel, thorough, and well-motivated.

Weaknesses
--------------
* The density of information in the results feels at times a bit overwhelming.
* Due to the volume of results presented, other relevant parts have been relegated to the appendix.

---

> ### Author Response · Authors · 2025-08-06
>
> We are deeply grateful for your appreciation of our work and for the insightful suggestions to enhance the presentation of the paper.
>
> ### Response to "Requested Changes"
>
> > The scaling law figures (figure 3-6) could be a bit scaled up for…
>
> Thank you for bringing this to our attention. In the revised version of our paper, we have scaled up those figures and moved the results for some LLM families (BLOOM and OPT) to the appendix.
>
> > It is also apparent to me that due to giving the presentation and…
>
> Thank you for this helpful suggestion. In the revised version of the paper, we have added more details about the datasets (Section 5.1) and experimental settings (Section 5.2). Additionally, in the discussion section (Section 7), we have included further insights into the aberrant behavior observed in the E2E and ViGGO datasets.
>
> > The same holds for the derivation in Appendix H - the result could…
> Thank you very much for pointing this out. In the revised version, we have introduced Section 8, which provides an abbreviated overview of our mathematical interpretation previously presented in the appendix.
>
> ### Response to "Other minor points"
>
> Those two minor points have already been corrected in the revised version of the paper.
>
> Thank you again for your thoughtful feedback. Please let us know if you have any further questions or require additional clarification; we would be glad to address them.

---

> > ### Comment · Reviewer_jHSa · 2025-08-06
> > **Response to Authors**
> >
> > Thank you very much for incorporating the requested changes!

---

### Review · Reviewer_7fDX · 2025-07-28

**Summary Of Contributions:**

This paper investigates the scaling laws of inconsistency in data-to-text generation across three major model classes: transformer decoders, transformer encoder-decoders, and state-space models. To conduct a rigorous analysis, the authors propose a framework called VaCScal, which evaluates predictive performance, goodness-of-fit, and performs comparative analysis between power-law and exponential scaling functions. Their findings suggest that the exponential function better captures the scaling behavior of inconsistency across all model classes and fine-tuning methods.

**Audience:**

Yes

**Claims And Evidence:**

Yes

**Requested Changes:**

To strengthen the paper, I recommend that the authors to address the weakness points, specifically,
1. extend their experiments to include more recent LLMs, and
2. larger models at 70B or above, also comapring with closed-model for strong baseline comparisons.
to ensure the findings are representative of current systems.
3. Additionally, since inconsistency in generation is often highly sensitive to prompt design, a thorough analysis of prompt formulation and its impact on consistency would be essential to validate the practical relevance of the conclusions.

**Strengths And Weaknesses:**

## Strengths

1. **Timely Topic:** The paper addresses an important problem in data-to-text generation, especially as many real-world systems transition from entity-based responses to free-form answers. Inconsistency in generation is a critical issue for LLM applications, and understanding how it scales with model size is valuable.

2. **Model Diversity:** By evaluating different model classes—transformer decoders, encoder-decoders, and state-space models—the paper provides broader insights beyond the commonly studied architectures. This sheds light on the potential of emerging LLM architectures.

3. **Statistical Rigor:** The use of well-established statistical tools for evaluating scaling behavior, goodness-of-fit, and functional comparisons (e.g., power-law vs. exponential) adds rigor to the analysis and supports the credibility of the findings.

---

## Weaknesses

1. **Outdated Model Choices:** The selected models (e.g., OPT, BLOOM, Pythia) are relatively outdated and may not reflect the behavior of current state-of-the-art LLMs. This limits the generalizability of the results.

2. **Limited Model Scale:** Most of the models studied are relatively small (≤13B). In practice, larger models (e.g., 70B or more) show significantly better consistency and reliability. Thus, the practical relevance of the findings is questionable.

3. **Lack of In-Depth Prompt Analysis:** Inconsistency in generation is often highly sensitive to prompt design. Many inconsistencies can be mitigated through better prompt engineering or few-shot instruction tuning. A thorough investigation into the interaction between prompting strategies and inconsistency is necessary to support the practical implications of the study's conclusions.

---

> ### Author Response · Authors · 2025-08-06
>
> We sincerely thank you for your thoughtful review of our paper and greatly appreciate your constructive feedback and suggestions.
>
>
> ### Response to "Weaknesses"
>
> > Outdated Model Choices
>
> In the revised version of our paper, we incorporated the Qwen2.5 family into our experiments. Qwen2.5 is a recently developed and widely adopted LLM family that has demonstrated outstanding performance on structured data and reasoning tasks. In the Qwen2.5 family, as with most of the other incorporated LLM families, we also observe a clear dominance of exponential scaling over power law scaling in modeling factual inconsistency.
>
> > Limited Model Scale
>
> In the incorporated Qwen2.5 family, we include seven models with parameter sizes ranging from 0.5B to 72B.
>
> > Lack of In-Depth Prompt Analysis
>
> We thank you for your valuable suggestion. The scaling studies presented in this paper primarily focus on fine-tuned LLM families rather than recent prompting mechanisms. This choice was made to ensure that our findings remain robust and are not influenced by variations in prompt design. As noted in the “Limitations and Future Scope” section, exploring the impact of prompting on scaling analyses for factual inconsistency in D2T remains an avenue for future work. Nevertheless, to provide greater clarity on our experimental setup, we have included a dedicated part Appendix A.5 of the revised paper detailing the prompt structure (used for supervised fine-tuning) and the linearization process for input D2T data.
>
> ### Response to "Requested Changes"
>
> We have already addressed this point in our response to the “Weaknesses” section.
>
>
> Thank you once again for your thoughtful feedback. Please let us know if you have any further questions or require additional clarification; we would be glad to address them.

---

### Author Response · Authors · 2025-08-06
**Revised Manuscript: Summary of Key Updates**

Dear Reviewers,

Thank you for your valuable feedback on our work. We have revised our manuscript to address your comments and have made the following key updates:

1. **Section 6 and Appendix D:** Incorporated the Qwen2.5 LLM family, including seven models ranging from 0.5B to 72B parameters.
2. **Appendix H:** Added scaling behavior analyses using full fine-tuning across multiple LLM families, including Qwen2.5.
3. **Appendix B:** Extended the human evaluation experiments beyond OPT.

In addition to these major updates, we have made several minor revisions, such as adding details on prompt design and input linearization (Appendix A.5) and reorganizing sections of the manuscript for improved clarity. All revised text is presented in "OrangeRed" color for easy reference.

Please let us know if you have any further questions or require additional clarification; we would be glad to address them.

---

### Author Response · Authors · 2025-10-08
**Acknowledgements**

We sincerely thank all the reviewers for their thorough evaluations and insightful suggestions. We are deeply appreciative of the Action Editor for ensuring a smooth and efficient publication process. On behalf of all authors, we extend our heartfelt gratitude to everyone involved for accepting our paper --- we are truly delighted and grateful.

---

### Decision · Action_Editor_jaKT · 2025-09-03

**Recommendation:** Accept as is

**Audience:**

Yes

**Audience Explanation:**

Certainly some members of the TMLR community would be interested in data-to-text generation with LLMs, and this paper focuses on an important asset, namely whether the LLM's generations are factually consistent with the data.

**Claims And Evidence:**

Yes

**Claims Explanation:**

All reviewers agreed that the experiments were convincing. Two reviewers asked for more "recent"/large models, and the authors added these results in their updated draft.